# From examination of natural events a proposal for risk mitigation of lahars by a cellular automata methodology: a case study for Vascún valley, Ecuador

Valeria Lupiano[1,2], Francesco Chidichimo[3], Guillermo Machado[4], Paolo Catelan[3,5], Lorena Molina[4], Claudia R. Calidonna[6], Salvatore Straface[3], Gino M. Crisci[2], Salvatore Di Gregorio[6,7]

[1]Consiglio Nazionale delle Ricerche, CNR-IRPI, Via Cavour 6, 87030 Rende, CS, Italy
[2]Department of Biology, Ecology, Earth Sciences, University of Calabria, Arcavacata, 87036 Rende, Italy
[3]Department of Environmental and Chemical Engineering, University of Calabria, Arcavacata, 87036 Rende, Italy
[4]Faculty of Engineering, National University of Chimborazo, Riobamba, Ecuador
[5]CEAA - Centro de Energías Alternativas y Ambiente,Escuela Superior Politécnica del Chimborazo, Riobamba, Ecuador
[6]ISAC - CNR, Lamezia Terme, Zona Industriale 88046 Lamezia Terme, CZ, Italy
[7]Department of Mathematics and Computer Science, University of Calabria, Arcavacata, 87036 Rende, Italy

*Correspondence to*: Francesco Chidichimo (francesco.chidichimo@unical.it)

**Abstract.** Lahars are erosive floods, mixtures of water and pyroclastic detritus, known for being the biggest environmental disaster and causing a large number of fatalities in the volcanic areas. Safety measures have been recently adopted in the threatened territories, by constructing retaining dams and embankments in key positions. More disastrous event could be generated by the difficulty of maintaining these works in efficiency and for the changed risk conditions originating from their presence and the effects of their functioning. LLUNPIY/3r, a version of the Cellular Automaton model LLUNPIY for lahars simulations is presented. The growing frequency of lahars in the Vascún Valley of Tungurahua Volcano (Ecuador), probably due to the effects of the climatic change, has recently produced smaller and less dangerous events, sometimes favoured by the collapse of ponds generated by small landslides. An investigation is here performed in order to reproduce such situations in controlled way by the use of LLUNPIY/3r simulations. Using precise field data, points are individuated where dams by backfills, easy to collapse, can produce the formation of ponds; LLUNPIY/3r simulations permit to project triggering of small lahars by minor rainfall events or to project in the case of larger rainfalls the anticipation of lahar detachment, avoiding simultaneous and dangerous confluence with other lahars.

## 1 Introduction

### 1.1 The problem of the lahars

Lahars are one of the most devastating phenomena as amount of fatalities in volcanic areas (Neall, 1976; Waythomas, 2014). They are flows, other than common stream flow, and consist of pyroclastic deposits mixed to water. Their physical properties (density, viscosity, consistency) are very similar to wet concrete not yet hardening (Vallance, 2000). This fluid, under steep

slopes conditions, is capable of reaching speeds up to100 km/h and distances up to 300 Km, it becomes solid when water is gradually released in flat areas, (Manville et al., 2013).

Lahars may be of primary type (or syneruptive) if directly related to volcanic eruptions, usually when glacier and/or snow are melt by pyroclastic or lava flows, they develop from mixing pyroclastic material with water as the tremendous 1985 Colombian

event of Nevado del Ruiz (Pierson et al., 1990); another case could occur when a large quantity of water is available by the breakout of a natural lake because of eruption (Manville, 2010).

Secondary lahars are produced when a large water quantity is available directly by extreme meteorological events or indirectly by the overflow of superficial water bodies. Pyroclastic deposits of previous eruptive activities are mobilized, e.g., the pyroclastic flows of Mt. Pinatubo, 1991 Philippines (Rodolfo et al., 1996).

Soil erosion with water inclusion along streams increases the volume of both primary and secondary lahars.

Two main triggering mechanisms are possible:

a) mobilization process related to pyroclastic sediments sometimes mixed with some exotic material (tephra): if the superficial water amount overcomes a water height threshold , related to features of pyroclastic stratum and soil slope, then the percolation can cause a detachment in the unconsolidated stratum;

b) erosion process  mainly depending on the redistribution of volcanic sediment along the slopes (Leavesley et al., 1989; Major et al., 2000): unconsolidated tephra ,swept away by flows, mixe with water and gradually enlarge their volume because of contribution of both sediments and waters (Barclay et al., 2007).

Different approaches were considered in literature for lahars modelling: empirical models (e.g. Schilling, 1998; Muñoz-Salinas et al., 2009) were developed, accounting mainly for some macro-observables phenomena; simplified hydrological and

rheological models that reduce the lahar behavior to a Newtonian-like behavior (e.g. O'brien, 1993; Costa, 1997); numerical methods approximating PDE (e.g. Pitman et al., 2003); Cellular Automata (CA) alternative methodology for lahar modelling will be exhibited later.

In many issues regarding complex systems, research was able of progress thanks to computer simulations, which allowed to develop multidisciplinary and transdisciplinary approaches, linked in part to the emergence of alternative computing

paradigms, such as CA (Toffoli, 1984, Chopard, 1998, Iovine et al., 2007).

### 1.2 Multicomponent or Macroscopic Cellular Automata

CA are both a parallel computational paradigm and an archetype for modelling "complex dynamical systems", that are extended in the space and can be described on the base of local interactions of their constituent parts. A homogeneous CA can be seen as a $d$-dimensional space, partitioned in cells of uniform size, each one embedding an identical input/output computing

device (a Finite State Automaton). Input for each cell is given by the states of the neighboring cells, where the neighborhood conditions are determined by a time and space invariant pattern. At the time $t$=0, cells are in arbitrary states (initial conditions) and the CA evolves by changing them simultaneously at discrete times (CA step), according to the transition function σ: $S^m{\rightarrow}S$, where $S$ is the finite set of the states and $m$ is the number of the neighbouring cells (Di Gregorio & Serra, 1999).

A short exemplification is given by the CA Majority: a two dimensions space is divided into square cells, so that the neighborhood of one square is given by the single element itself together with the eight surrounding it. Their states are blue (0) and red (1), and are added together within the neighborhood by the transition function. If the sum is more than 4 (the majority of neighbors is red), the next state of the cell will be red otherwise it will be blue. Sometimes the system evolves from

an initial distribution of reds and blues in a complex way, originating local points of expansion of colors (Toffoli, 1984).

When complex macroscopic dynamical systems, like "surface flows" phenomena (lahars, debris flows, snow avalanche, lava flows, and pyroclastic flows), are modelled, the previous definitions are insufficient, and Multicomponent or Macroscopic CA (MCA) adopt the following extensions.

The abstract CA must be related univocally to the real phenomenon in its dynamics, each cell has to correspond to a portion

of the space or surface (of the territory $T$) where the phenomenon evolves, so the time corresponding to a step of the transition function has to be set, the size of the cell has to be specified, e.g. by the length of its edge. These constant values in time and space are called global parameters. $P$ is the set of global parameters, it includes both physical and empirical parameters. The choice of some parameters is imposed, where possible, by the precision that must be achieved in the simulation, e.g. cell dimension. The value of some other parameters is deduced instead by the physical features of the phenomenon, e.g. the

parameter related to energy dissipation by turbulence. In these cases, the initial physically sounding value considered at the beginning of validation, is corrected by attempts in the phase of model validation on the base of the simulation quality, depending on the comparison between real event and simulation results. A methodology, based on Genetic Algorithms, was usually used for calibrating the parameters of our CA models of surface flows (Iovine et al., 2005).

Each characteristic, relevant to the evolution of the system and relative to the space portion corresponding to the cell, is

individuated as a substate. The finite set $Q$ of the states is given by the Cartesian product of the substates: $Q=Q_1 \times Q_2 \times ...... \times Q_n$. Examples of a lahar model substates related to the part of territory corresponding to a cell are: the average altitude (substate altitude), the thickness of the lahar (substate lahar thickness), the depth of erodible (unconsolidated) pyroclastic stratum (substate pyroclastic stratum depth). The dynamics of the phenomenon is expressed by the variation of the substates values in the successive steps of simulation for each cell. Considering that the features related to the third dimension may be expressed

in terms of substates, it is possible to develop two dimensions models operating three-dimensionally in fact (Avolio et al., 2012).

MCA have to account for phenomena, whose dynamics involves more interacting processes, sometime of different nature, e.g., loss of lahar energy because of erosion of the unconsolidated pyroclastic stratum of the "cell", loss of energy of the lahar in the "cell" caused by its turbolence. These interacting processes compose the transition function and are called "elementary"

processes of the CA. They are computed sequentially, involving the update of the MCA substates.

The last extension of MCA are the "external influences", that account for input from the "external world" not depending on local interactions (that cannot be reduced to local interactions) occurring at some cells of the CA, e.g., the external influence "lava alimentation at the vents" is applied, at each step, only to those cells corresponding to the locations where the vents actually are. The value of the substate "lava quantity" is updated by adding the amount that is considered to be discharged (in

the case of simulation of a real event) or that is supposed to be discharged (in the case of simulation of a conjectured event) in the cell during the time step (Di Gregorio and Serra, 1999).

Simulations of flow-like landslides were performed by several versions of the MCA model SCIDDICA since 1987 for both subaerial and subaqueous debris/granular/mud flows (e.g., Barca et al., 1987; Avolio et al. 2008; Mazzanti et al., 2010; Avolio

et al. 2013; Lupiano et al., 2014; Lupiano et al., 2015a; Lupiano et al., 2015b; Lupiano et al., 2015c; Lupiano et al., 2017). Simulations of primary and secondary lahars were performed by the MCA model LLUNPIY (Machado et al., 2014; Machado et al., 2015a; Machado et al., 2015b; Chidichimo et al., 2016).

LLUNPIY, SCIDDICA-SS3 and SCIDDICA-SS2 are our most advanced models (in the sense that they include the features of the previous models plus other new ones) for simulating flow-like landslides and lahars. Unlike other models, that were

used in lahar simulations (LAHARZ (e.g., Schilling, 1998; Muñoz-Salinas et al., 2009), TITAN2D (e.g., Sheridan, 2005; Williams, 2008; Córdoba et al., 2014)), they allow the implementation of the erosion process.

## 1.3 Strategies of risk mitigation for lahars

Reliable simulation tools are very important in order to develop risk mitigation strategies and to test them in different conditions. Such instruments have to be used with extreme caution, because the complex problem of lahar hazard must be

studied with an interdisciplinary approach (Lane et al., 2003; Leung et al., 2003), e.g., mitigation measures which involve engineered protection structures could modify hazard conditions in the time and could increase the disaster risk as better specified below.

Beside tools of early warning, which could work only partially, beside temporary or definitive land evacuation which could involve a strong social impact and economic destitution, security measures have been adopted in volcanic territory, by

constructing retaining dams, embankments, walls, dykes, levees, reservoirs in key positions for containing and deviating possible lahars (Scott, 1989; Verstappen, 1992; Aguilera, 2003; Künzler et al., 2012; Carey et al., 2012). This solution could involve a strong environmental impact: it is difficult to guarantee the constant efficiency of these works, and their presence, together with the effects of their functioning could severally change the risk conditions (Janda et al., 1981, 1996; Scott, 1989; Procter et al., 2010, Shreve & Kelman, 2014, Wisner et al., 2012).

More in general a short paper of Kelman (2007) underline synthetically that: "Despite decades of evidence from research and practice demonstrating that reliance on structural approaches increases disaster risk over the long-term, structural approaches are frequently preferred without properly considering complementary or alternative measures. Examples of structural approaches are walls, dams, dykes, levees, and reservoirs. While they do provide some benefits, decisions to implement them and nothing else are usually made by emphasizing the short-term benefits and discounting the long-term costs".

The growing frequency of lahars in the area of Vascún Valley of Tungurahua Volcano, Ecuador has recently produced smaller (shorter accumulation periods) and therefore less dangerous events (Mothes and Vallance, 2015). Moreover, small landslides, forming natural dams with temporary ponds, could easily trigger lahars by collapsing because of rainfalls; it sometime happens, e.g. the IGEPN (Instituto Geofísico Escuela Politecnica Nacional, Quito, Ecuador) reported such a case of August, 23 2008

(2008a; 2008b). These extraordinary combinations of events gave birth to the idea of using the overabundant pyroclastic material, available on site, to create easy to collapse artificial dams. The dam breakdown is obtained through the appropriate sizing of the cross section of the structure which is designed to fail at the achievement of a specified water level. This goal is reached through the implementation of an ad hoc numerical model, based on the Finite Element Method (FEM), for the stability

analysis of the dam slopes.

For such points, minor rainfall events can produce small lahars, major events will anticipate the lahar detachment, avoiding simultaneous confluence with other lahars. The control of the dams collapse could permit in various situations many combinations for a controlled triggering of lahars, in order to mitigate the risk.

The next chapter is devoted to the geological description of the Vascún Valley, the third chapter introduces LLUNPIY/3r, the

MCA model for simulating primary and secondary lahars, together with its validation in simulating some significant lahars of Vascún Valley. In the fourth chapter, the building of dams, easy to collapse, is considered, then their favorable locations are hypothesized for a controlled triggering of lahars, the effects of which are simulated for possible events; at the end, conclusions and comments.

## 2 Geological setting of Vascún valley

Tungurahua is one of the most active and dangerous volcanoes in the Ecuadorian Andes (Cordillera Oriental) on the inside of the Sangay National Park; its summit, 5023 m.a.s.l., is positioned at longitude 78° 27' W, latitude 01° 28' S. It is a stratovolcano, whose evolution involved the succession of three major volcanic edifices (Tungurahua I, Tungurahua II, Tungurahua III) since the mid-Pleistocene over a basement of metamorphic rocks. Historical eruptions have all originated from the summit crater. The main events of eruptive activity occurred between: 1640–1641, 1773–1777, 1886–1888, 1916–

1918 and from 1999 until the present (Hall et al., 1999; Ramón, 2009; Biggs et al., 2010). The average of eruptions in the last two thousand years is once per century according to the detailed studies of Le Pennec et al. (2008). They have been accompanied by strong explosions and sometimes by pyroclastic and lava flows that reached populated areas at the volcano's base. The formation of rain-induced lahar is also a cause of danger. Approximately 32,000 people live within the higher risk areas, mainly in rural villages and in the touristic (thermal springs) town of Baños de Agua Santa (Mothes et al., 2015). Baños

de Agua Santa (1800 m.a.s.l.) is only 8 Km as the crow flies away from the summit; 30 Km to NW, 30 km to SW and 140 km to N are the distances respectively from the towns of Ambato, Riobamba, and from the capital Quito.

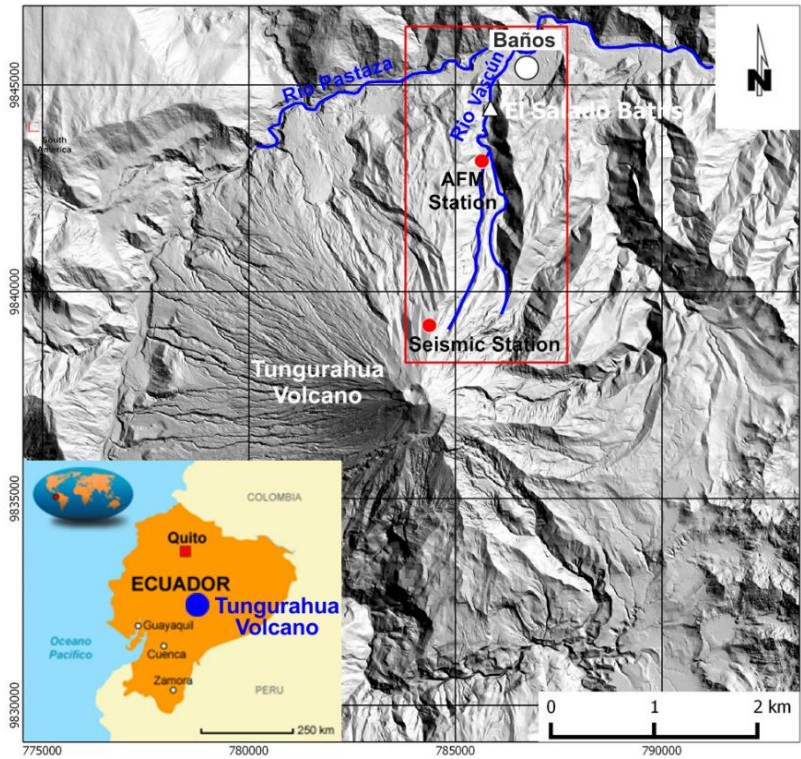

**Figure 1: Tungurahua Volcano. The Vascún Valley is inside the red box.**

The small glacier of Tungurahua volcano is reduced both for the global warming phenomenon and for the intensification of the volcanic activity after the 1999, therefore today's snow cover is completely negligible toward primary lahars generation.

Lahars triggering, with significant frequency and magnitude, is hence subordinated both to the intensity and duration of the rainfalls and the available quantity of fresh material (cumulated unconsolidated pyroclastic material from volcanic eruption) along the slopes and within the principal canyons (Quebradas of the Rio Vascún, Juive Grade-La Pampa Valley, Achupashal Quebrada) and to a lesser extent to other factors. Almost all the lahars are confined to the canyons and converge into Pastaza River (Mothes and Vallance, 2015).

The eruptive activity of Tungurahua volcano, during the last years, has generated a greater availability of pyroclastic debris that is periodically remobilized from atmospheric phenomena, often not particularly violent but prolonged for several days. Between 2000 and 2011, around 900 rain-induced lahars were triggered (Mothes and Vallance, 2015).

Generally, lahars magnitude is small and, consequently, causes limited damage. Precious data were supplied by the Instituto Geofísico Escuela Politecnica Nacional, Quito, Ecuador (IGEPN) and its survey stations around the country

(https://www.igepn.edu.ec/), it is the maximum authority for various volcanic hazards and earthquakes. IGEPN and its Acoustic Flow Monitor (AFM) station detect the passage of the majority of secondary lahars (e.g.: IGEPN, 2005; 2008a; 2008b), while many others are traced by the Observatory of the Volcano Tungurahua (OVT), which is situated 13 km to the north-northwest of the crater, also with the observation contribution of local volunteers (vigias).

## 2.1 The 2005 and 2008 secondary lahars of Vascún Valley

Rio Vascún gave its name to the Valley on the NE flank of Tungurahua Volcano, it flows into the Rio Pastaza (Fig.1). The valley slopes exceed 35°-40°in the first steepest 3 km, while range from 20° to 6° in the last less steeply 2-3 Km. Furthermore, the path of Rio Vascún is extremely sinuous in the upper 1-2 km where the river is characterized by a succession of tight 90° bends.

The Vascún Valley is highly susceptible to lahar flows that threaten the nearby populated areas which were inundated several times in the past years: the town of Baños, that extend in part of the depositional area, was affected several times, the thermal structure "El Salado", that is located along the river banks, had been repeatedly damaged by passage of lahars (Fig. 2b).

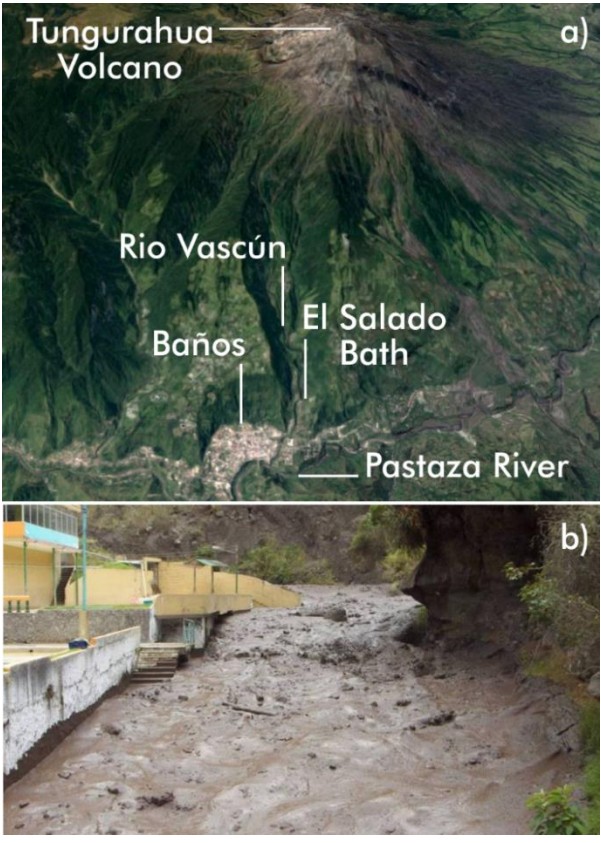

**Figure 2: a) Google Earth view of Tungurahua Volcano with indication of main localities of the study area; b) 2005 lahar at thermal structure of "El Salado".**

For LLUNPIY/3r calibration and validation two events were taken into consideration, which took place respectively in 2005 and 2008. The dynamic of the events is briefly described below.

On February 12, 2005, heavy rainfalls caused the remobilization of ash fall deposits, generating lahars in the Rio Vascún Valley. The mean velocity, estimated on the base of data recorded by the alert instrumentation, varied between 7 m/s and 3 m/s (Williams et al., 2008) according to the morphological characteristics of valley crossed sector. The lahar volume, measured

by the staff of the AFM station, was estimated at approximately 70000 m$^3$, while a subsequent investigation, carried out by IGEPN (2005), estimated it to be 55000 m$^3$. The lahar crossed the valley for about 10 km flooding El Salado Baths during the passage, and then reached the Pastaza River. The chronicle of the event is taken from the work of Williams et al. 2008 and we refer to their data and simulations for comparison with our simulation.

The 2008 lahar had a different dynamic. On August 13, a small landslide produced a natural dam along Rio Vascún at an elevation of about 2200 m a.s.l. The dam originated a pond with a length of 100 m, a depth of 3 m and a width of 20 m. After heavy rains on August 22, the dam collapsed and generated a lahar. The flow velocity was estimated in 15 m/s with a flow rate of 1120 m$^3$/s and an average height of 4 m (IGEPN, 2008a and 2008b), this flow rate is ten times greater than that recorded in 2005 event (IGEPN, 2005, Williams et al., 2008). The lahar reached in 5 minutes El Salado and devastated the pools of the
thermal spa, afterwards it destroyed some houses of Las Ilusiones (a village of Baños district) further downstream.

**3 LLUNPIY/3r model for lahar simulation**

LLUNPIY (Lahar modelling by Local rules based on an UNderlying PIck of Yoked processes, "llunp'iy" means flood in the Quechua language) is a model for simulating secondary and primary lahars according to MCA methodology applied to complex system, whose evolution may be mainly specified in terms of local interaction. MCA features of SCIDDICA-SS3 (Avolio et
al., 2013) and SCIDDICA-SS2 (Avolio et al., 2008; Lupiano et al., 2016; Lupiano et al., 2017) are inherited by LLUNPIY; LLUNPIY for secondary lahars is extensively defined in Machado et al. (2015b), here only the features of the model, that were applied in the study cases, are reported (reduced version LLUNPIY/3r from SCIDDICA-SS2). No external influences were considered. The LLUNPIY/3r simulation starts considering data related to the altitude (value of the "altitude" substate, see Chapter 1.2), to the depth of erodible pyroclastic stratum (value of the "soil of the cell" substate, see Chapter 1.2); to the lahar
thickness (value of the "thickness of the lahar " substate, see Chapter 1.2), for each cell.

A reliable reconstruction of the first phase of a real event of lahar permits to set an "initial" moment, where it is possible to deduce the thickness of lahar in the territory, these data constitute the values of the substate "thickness of the lahar" in the first step of the simulation. In the case of the simulation of a lahar produced by the collapse of a dam holding a given water volume, the thickness of lahar is deduced by the mixing of pond water with the dam material and part of the unconsolidated pyroclastic
stratum below. Note that the simulated lahar events, occurred in the Vascun Valley, do not involve the very first phase of water percolation and detachment subsequent to water inclusion (Machado 2015), since the collapse of temporary pond is abrupt. In the cases of past event, data permitted simulation of the phenomenon just in the phase of lahar. Furthermore in the simulation of real and hypothesized events, all the lahars end into the Rio Pastaza, so the last phase of lahar deposition is omitted and the viscosity of lahar may be considered constant for these particular cases.

**3.1 Introduction to the LLUNPIY/3r version**

The following quintuple defines the two-dimensions (with hexagonal cells) MCA model LLUNPIY/3r

$$<T, P, N, Q, \sigma>$$

where:

- $T = \{(x, y) \mid x, y \in \mathbb{N}, 0 \le x \le l_x, 0 \le y \le l_y\}$ is the set of hexagonal cells, that tessellate the territory, where the phenomenon evolves; the cells are individuated by the points with integer co-ordinates (Fig.3, left) of their centers; $\mathbb{N}$ is the set of the natural numbers.

- $P$ is the set of both the empirical and physical global parameters, they are related to the global common features of the phenomenon (Table 1);

- $N = <(1,-1), (0,-1) (-1,0), (-1,1), (0,1), (1,0), (0,0) >$ the neighborhood, identifies the geometrical pattern of cells (Fig.3a), which influence the state change of the "central" cell; index 0 is assigned to the central cell and indexes 1,..,6 are assigned to the six adjacent cells (Fig.3b); #$N$=7. The sum of indexes of opposite directions is always 7.

- $Q$ is the finite set of states of the finite states automaton, incorporated in the cell; it is specified in terms of substates as their Cartesian product (Table 2).

- $\sigma: Q^{\#N} \rightarrow Q$ is the deterministic transition function for each cell in $T$, the following "elementary" processes compose $\sigma$, they account for the overall dynamics of the phenomenon:

  - $\tau_{mob}$, effects of mobilization
  - $\tau_{lo}$, lahar outflows
  - $\tau_{te}$, effect of turbulence
  - $\tau_{fc}$, composition of flows

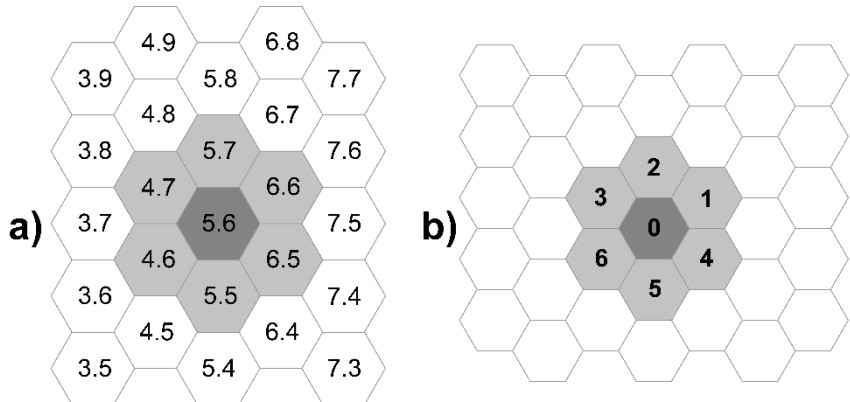

**Figure 3: a) The neighborhood of cell (5,6); b) neighborhood indexes.**

Physical parameters regard physical quantities that are used in equations of the transition function and correspond to values adopted in the implementation of the model (e.g. cell apothem $p_a$ that depends on several factors, data precision, insuperable approximation limits related to specific features of the phenomenon) or values as the temporal correspondence of a CA step

$p_t$, that must account that $p_a/p_t > v_{mx}$ where $v_{mx}$ is the maximum possible velocity of flows during the development of the phenomenon. In other words the shift of a flow in a CA step has not to overcome the neighborhood. All the parameters, except $p_a$ and $p_t$, are empirically set in the phase of model validation by the simulation quality, initial values of parameters were deduced by the physical features of the phenomenon, e.g. a very slow lahar would emerge unbelievably in simulation by largest

values of $p_{cf}$, the coefficient of friction and $p_{dt}$, the energy dissipation due to turbulence

## 3.2 The elementary processes of LLUNPIY/3r

In the following an outline of the transition function, with a description of the elementary processes updating the substates, will be provided. The complete execution of all the elementary processes accomplishes a step of the LLUNPIY/3r. Neighborhood index between square brackets, following substate specification, indicates the corresponding cell of the

neighborhood. $\Delta Q_S$ indicates variation of the sub-state $Q_S$. $Q_S'$ indicates the new value of the substate $Q_S$., $Q_S' = Q_S + \Delta Q_s$ . In the case of external and internal flows, the cell, to which the flow is directed, is specified in the substate, inserting a subscript, which precedes it, e.g. $_2Q_E[1]$, i.e. the external flow of the neighbor with index 1 toward its neighbor with index 2.

*Pyroclastic cover mobilization*

Soil features together with the quantity of water content determine a value $p_{tm}$ of mobilization threshold to be compared with the kinetic head $Q_{KH}$ associated to lahar debris $Q_{LT}$; when $Q_{KH} > p_{tm}$, the pyroclastic cover $Q_D$ is eroded, the lahar thickness $Q_{LT}$ augments and altitude $Q_A$ diminishes according to the following empirical formula, that turned out to be valid in different models of debris flow e.g. (Avolio et al., 2008), snow avalanche (Avolio et al., 2017) and primary and secondary lahars e.g. (Machado, 2015).

$$-\Delta Q_D = \Delta Q_{LT} = -\Delta Q_A = (Q_{KH} - p_{tm})\, p_{pe} \,, \tag{1}$$

There is correspondingly a dissipation of energy, proportional to the depth of erosion, it is specified by a decrease of kinetic head $Q_{KH}$ according to the following formula:

$$-\Delta Q_{KH} = (Q_{KH} - p_{tm})\, p_{de} \,, \tag{2}$$

*Effect of turbulence*

A loss of kinetic head occurs by turbulence at each LLUNPIY/3r step according to the following equation:

$$-\Delta Q_{KH} = p_{dt} Q_{KH} \tag{3}$$

where $p_{dt}$ is an empirical parameter that accounts of the turbulence kinetic energy (Lander & Spalding, 1973), such parameter is referred in LLUNPIY/3r to the substate $Q_{KH}$, that is directly related to the kinetic energy.

*Lahar outflows*

$f[i], 1 \le i \le 6$ , specify the outflows from the central cell toward the adjacent cell, $f[0]$ is the part remaining in the central cell. They are computed in two steps: application of the Algorithm of the Minimization of Differences, AMD (Avolio *et al.*, 2012; Di Gregorio and Serra, 1999) to the "heights" in the neighborhood of the central cell and calculation of the shift of the $f[i], 1 \le i \le 6$ (Avolio *et al.*, 2013).

AMD application computes the outflows $f[i], 1 \le i \le 6$, which minimize the "height" differences in the neighborhood (equation 7). An alteration of height values is introduced in the central cell for taking into account the outflow run-up; furthermore the viscosity is modelled by an adherence "$adh$" term, the lahar quantity, that cannot leave the central cell. It varies between the two extreme values $adh1$ and $adh2$, which depend on the composition of the pyroclastic debris at the maximum and minimum water content (Machado, 2015c).

This "adherence" method was initially used for modelling lava flows by CA, in order to manage the continuous variation of viscosity by cooling of lava e.g., (Avolio et al., 2006). The approximation to account for viscosity inside a CA context can be intuitively explained as follows. Without bring into play a system in which innumerable fluid layers flow one over the other, at most two layers are considered. The first layer, whose maximum thickness ($adh$) is determined by the coefficient of viscosity, can not move, if the thickness of fluid *th* overcomes *adh*, a second layer with thickness *th-adh* is considered to slide on the

first one with a friction coefficient related to viscosity.

$$h[0] = Q_A[0] + Q_{KH}[0] + adh ,\tag{4}$$

$$h[i] = Q_A[i] + Q_{LT}[i], (1 \le i \le 6) ,\tag{5}$$

$$q = Q_{LT}[0] - adh = \sum_{0 \le i \le 6} f[i]\tag{6}$$

$$\sum_{\{(i,j)/0 \le i < j \le 6\}}(|(h[i]+f[i]) - (h[j]+f[j])|)\tag{7}$$

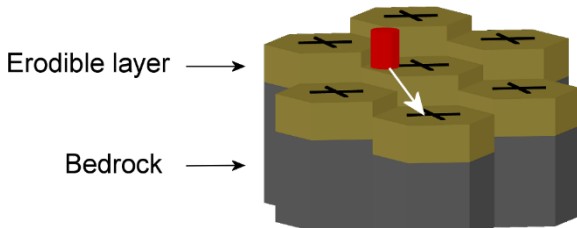

**Figure 4: Outflow direction from central cell to the center of an adjacent cell in 3-dimensions.**

Each moving quantity $f[i]$ (where in the following $1 \le i \le 6$) may be considered as a "cylinder", at first entirely inside the cell, having a center of mass with co-ordinates $Q_X[0]$ and $Q_Y[0]$ and with the maximum possible radius.

The shift "$sh[i]$" of $f[i]$ is calculated according to the following formulae, where the movement of the mass center is specified as the mass movement on a constant slope with a constant coefficient of friction $p_{cf}$. The movement of $f[i]$ is directed towards the center of cell $i$ with co-ordinates $Q_X[i]$ and $Q_Y[i]$, considering the slope angle $\theta[i]$ (Fig.4).

$$sh[i] = v \cdot p_t + g(\sin\theta[i] - p_{cf}\cos\theta[i]) p_t^2/2 , \quad (1 \leq i \leq 6) , \tag{8}$$

with "$g$" the acceleration of gravity, "$v$" the initial velocity:

$$v = \sqrt{(2g \cdot Q_{KH}[0])} \tag{9}$$

There are three possible outcomes: if the shifted cylinder is completely inside (outside) the central cell, there is only an internal (external) outflow, otherwise two cylinders form with mass center corresponding to the mass center of the internal outflow and of the external outflow. The new positions ( $_iQ_{EX}[0]$ and $_iQ_{EY}[0]$ and $_iQ_{IX}[0]$ and $_iQ_{IY}[0]$) of external and internal

outflow $_iQ_E[0]$ and $_iQ_I[0]$ account also for the variations of kinetic head $_iQ_{KHE}[0]$ and $_iQ_{KHI}[0]$..

*Flows Composition*

Execution of the elementary process "lahar outflows" involves an updating of substates $Q_{LT}$, $Q_{KH}$, $Q_X$, $Q_Y$ by the elementary process "Flows Composition". It accounts for the variation of lahar content of the cell, i.e. variation of $Q_{LT}$ and corresponding

variation of $Q_{KH}$, $Q_X$, $Q_Y$ that is determined by the external outflows $Q_E$, they represent inflows for the cells to which they are directed. Internal outflows $Q_I$, determine just a shift inside the cell, with variations of the substates $Q_{KH}$, $Q_X$, $Q_Y$ (Machado et al., 2015).

The value of the substate $Q_{LT}$ at the next step is given by the its previous value minus the losses determined by the outflows $_iQ_E[0]$ from the cell (normalized to a thickness) plus the contributions determined by the inflows $_{7-i}Q_E[i]$ from the neighbors.

$$Q'_{LT}[0] = Q_{LT}[0] + \sum_{i=1}^{6}(_{7-i}Q_E[i] - _iQ_E[0])$$

The other substates change correspondingly, considering a weighted average

$$Q'_{KH}[0] = \frac{Q_{KH}[0] \cdot Q_{LT}[0] + \sum_{i=1}^{6}(_{7-i}Q_{KHE}[i] \cdot _{7-i}Q_E[i] - _iQ_{KHE}[0] \cdot _iQ_E[0])}{Q_{LT}[0] + \sum_{i=1}^{6}(_{7-i}Q_E[i] - _iQ_E[0])}$$

The shifts both of the external flows and internal flows have to be considered for the new values of co-ordinates substates.

$$Q'_X[0] = \frac{Q_X[0] \cdot (Q_{LT}[0] - \sum_{i=1}^{6}(_iQ_I[0])) + \sum_{i=1}^{6}(_{7-i}Q_{EX}[i] \cdot _{7-i}Q_E[i] + _iQ_{IX}[0] \cdot _iQ_I[0] - _iQ_{EX}[0] \cdot _iQ_E[0])}{Q_{LT}[0] + \sum_{i=1}^{6}(_{7-i}Q_E[i] - _iQ_E[0])}$$

$$Q'_Y[0] = \frac{Q_Y[0] \cdot (Q_{LT}[0] - \sum_{i=1}^{6}(_iQ_I[0])) + \sum_{i=1}^{6}(_{7-i}Q_{EY}[i] \cdot _{7-i}Q_E[i] + _iQ_{IY}[0] \cdot _iQ_I[0] - _iQ_{EY}[0] \cdot _iQ_E[0])}{Q_{LT}[0] + \sum_{i=1}^{6}(_{7-i}Q_E[i] - _iQ_E[0])}$$

### 3.3 LLUNPIY calibration and validation

We selected the 2005 and 2008 lahars of Vascún Valley respectively for LLUNPIY/3r version calibration and validation. Available data, although incomplete, of the flood phase (Machado et al., 2015b) seemed promising in order to obtain reliable simulations. In fact data of different sources were carefully compared and analyzed (Williams et al., 2008; IGEPN, 2008) in order to reconstruct as accurately as possible the two events (Machado et al., 2014a and 2014b).

The use of simulation tools (from the cellular automata model LLUNPIY) needs detailed field data: DEM, depth of erodible pyroclastic stratum. It implies accurate geological investigations, including subsoil tomographies. Geophysical surveys allow to individuate points, where dams by backfills, easy to collapse, can enable the formation of ponds, whose breakdown can trigger a lahar (Machado, 2015c; Chidichimo et al., 2016).

The simulation of 2005 event is based on a Digital Elevation Model (DEM) with 1m cell size (supplied to us by Dr. Gustavo Cordoba), while the 2008 lahar was performed with a DEM of 5m cell size (supplied by IGEPN). In both cases a uniform thickness of 5 m was imposed for detrital cover, because detailed surveys were not available. This introduces a series of approximations that negatively influence the results of simulations. Such approximations can be reduced by an opportune survey of field data, e.g. by soil tomographies, MASW, coring, etc.

The same set of LLUNPIY/3r parameters was used in the two cases except for the parameter of progressive erosion ($p_{pe}$) because of different percentages of water in the soil. The 2005 event was triggered in a higher and very slope zone of Rio Vascún, when the water concentration in the soil reached critical values due to rainfall. The 2008 event was dissimilar, because the breaking of a temporary pond suddenly released a larger water quantity (in comparison with 2005 case) with strong turbulence, whose effects correspond to a higher value of the parameter of progressive erosion (Machado et al., 2015b).

The results of the simulations of 2008 event (Machado et al. 2015b) are extensively reported in this study since this event, as just said before, was caused by the same type of phenomenon, whose development, we want to forecast. The reliability of the simulation results, in comparison with the real event, has led to confide in the goodness of the method, the new simulations were performed with the same data precision and the same values of parameters.

Simulations of 2005 event were limited by the partial data field and DTM information: a stretch of about 2.3 km, from elevation 2150 m a.s.l. (about 850 m upstream of El Salado Bath) to elevation 1900 m a.s.l. (in correspondence of Pastaza River) was considered. The area, where the simulation starts, does not concern the detachment phase that occurs 8 km upriver. A kind of detachment, where an initial velocity of 7 m/s was imposed to lahar, was considered in order to express the first arrival of lahar flows. An equivalent fluid approach was adopted, because precise data about water flows are not available. Therefore, bulking must account not only for erodible layer, but for water inclusion. The total mass is inclusive of the water one. This generates a discrepancy between the lahar volume, measured on the deposit, and the "fluid" lahar volume including water to be loss in the last part of the event.

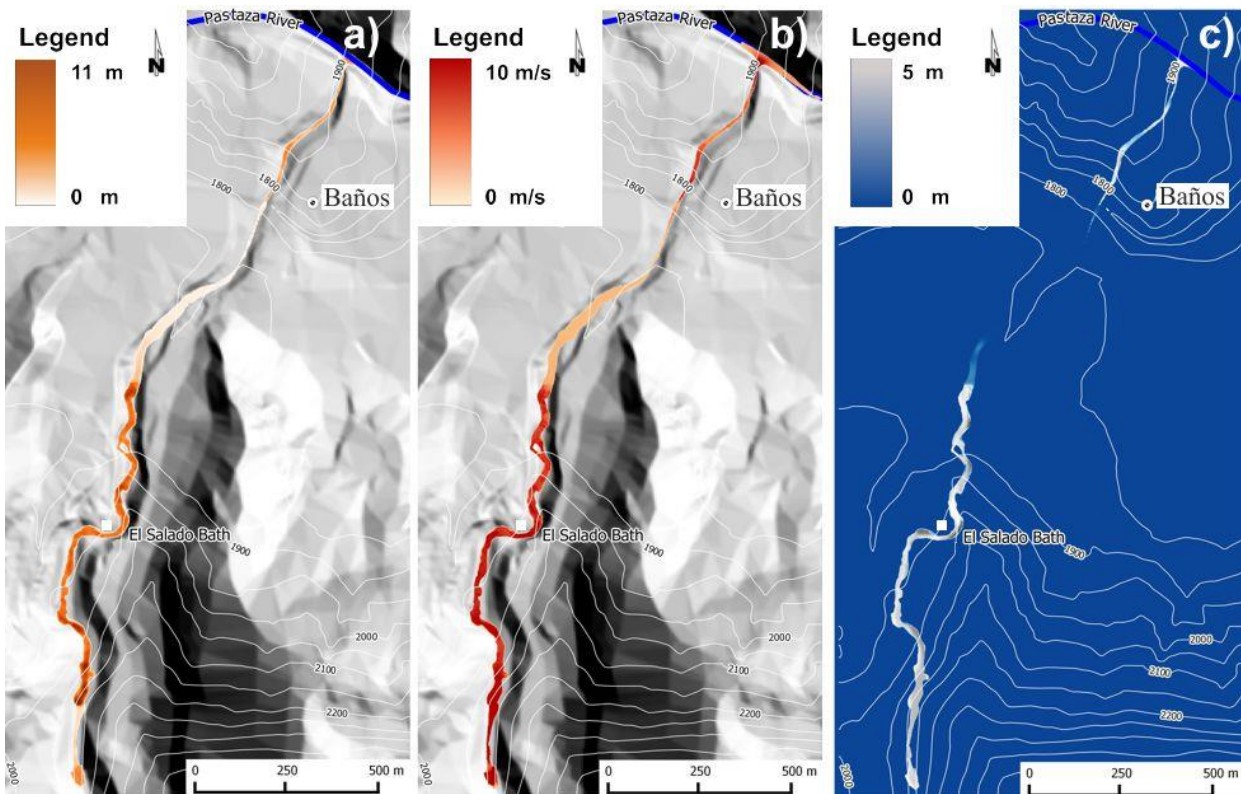

**Figure 5: (a) Maximum thickness, (b) maximum velocity, and (c) erosion depth, in 2005 simulated event.**

Fig. 5 shows the simulation developed with LLUNPIY in the considered sector. In particular, the maximum debris thickness values, which were reached by the lahar in simulation, are reported in Fig. 5a. Maximum velocities, reached by simulated flows (Fig. 5b), are high in steeper areas (the expected result) and gradually decrease at the downstream outlet. A velocity increase occurs south of Baños, probably because of the higher gradient of the river bed. Erosion has a trend similar to that of velocity (Fig. 5c). Table 3 synthesizes values of Fig. 5 and compares such data with IGEPN field data, reported in IGEPN 2005, and with simulation performed by Titan2D (Williams et al., 2008). Such field data are obviously partial for the complete development of catastrophic phenomenon, but extremely precious for the comparison with our simulations. Observation data are not sufficient for a precise comparison with the simulation paths. Furthermore, the lahar starts with null velocity in the simulation of (Williams et al., 2008), while LLUNPIY simulations start with 7 m/s velocities. The difference for total eroded mass rises from the lost water volume that was not possible to be considered in measurements.

The simulation of the 2008 lahar is shown in Fig. 6: the flow speed has reached values up to 20 m/s in many areas of the valley, and the eroded material has resulted in a volume of about 970000 m$^3$. The maximum height obtained in the simulated flow (Fig. 6a), in some sectors where the valley is particularly narrow, is 22 m , while the estimated average value by IGPEN (2008a and 2008b) is 4 m.

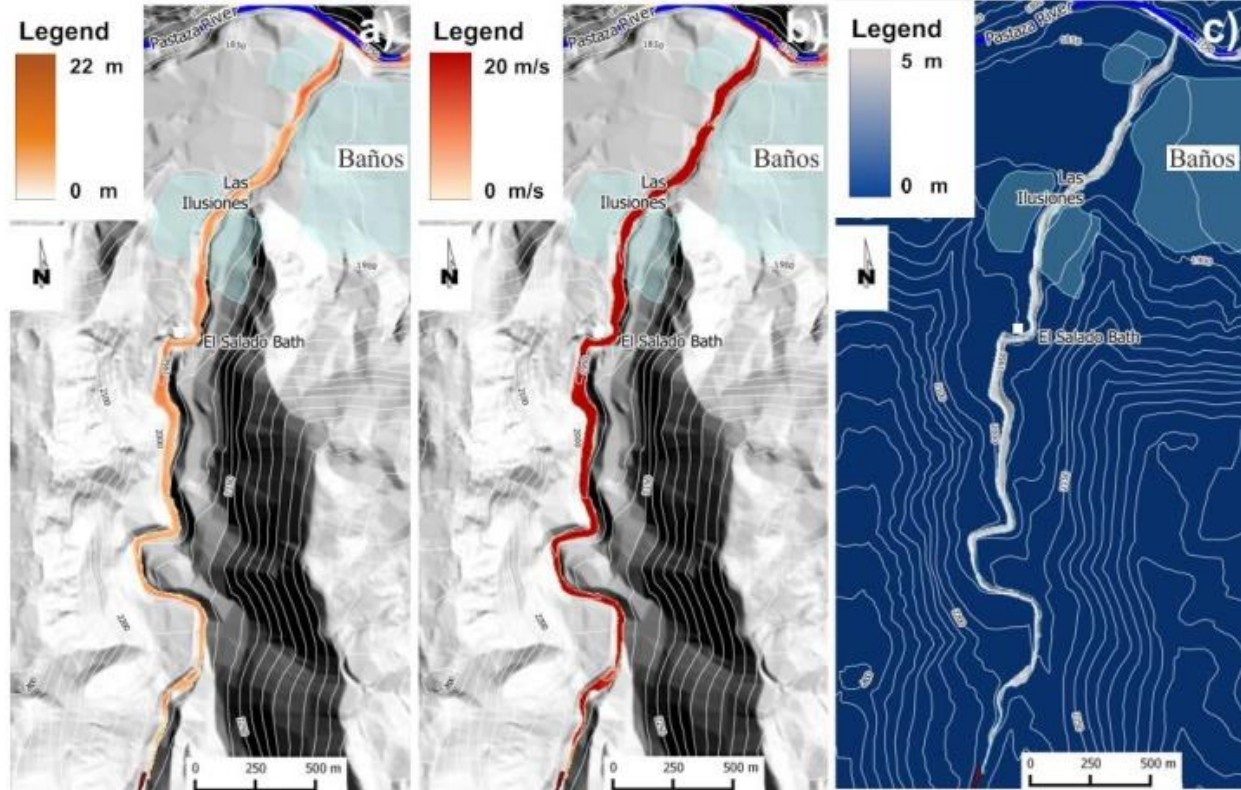

**Figure 6: 2008 simulated event. a) Maximum thickness, b) Maximum velocity, and c) erosion depth.**

Table 4 compares some data of simulation by LLUNPY with corresponding field data of IGEPN (2008). It is possible to note that the data deriving from the simulations are not very different from the known measured ones. The flow velocity of 15 m/s represents an estimated value, not a measured one. These results demonstrate that LLUNPIY/3r is a reliable model, given that the simulations are based on incomplete and sometime very approximate data concerning the pre-event and post-event. Furthermore the inevitable errors, in the records related to this event, have to be considered. Therefore an extension of LLUNPIY/3r is promising in order to introduce secondary features of the phenomenon to be tested. Simulations reproduce satisfactorily the overall dynamics of the events: there is a good matching between real and simulated lahar path, velocity and height of detrital flow. Note that different approaches always obtain excellent results about the path because the lahar is canalized by steep faces.

## 4 Lahar triggering and effects

### 4.1 Building rudimental dams easy to collapse

Ponds form along watercourses in volcanic areas, when landslides of volcanic deposits, which are originated by pyroclastic flows and lahars, act as a dam by obstructing the stream bed. The most frequent cause of a breakout of such natural ponds is

the overflow of water across the newly formed dam during violent rainfalls and subsequent erosion and rapid cutting down into the loose rock debris. The classification of the "Glossary of Meteorology" of the American Meteorological Society for rainfall intensity (Glickman, 2000) is here adopted according to the rate of precipitation $R_p$: measured in mm h$^{-1}$:

Light rain: $R_p < 2.5$ mm h$^{-1}$;

Moderate rain: $2.5$ mm h$^{-1} \leq R_p < 10$ mm h$^{-1}$;

Heavy rain: $10$ mm h$^{-1} \leq R_p \leq 50$ mm h$^{-1}$;

Violent rain: $R_p > 50$ mm h$^{-1}$.

Dam collapse occurs when instability conditions arise in the downstream slope. By eroding the obstruction and flowing downstream along the river bed, the initial surge of water will incorporate a dangerous volume of sediments. This can easily

produce lahars with possible devastating effects for settlements in their path (Leung et al., 2003).

Temporary dams with a similar (but controlled) behavior can be designed and built at low cost by local backfills in order to allow the outflow of streams produced by regular rainfall events. This result is achieved by properly dimensioning the embankment according to a stability analysis. The latter is made by comparing the forces tending to cause movement of the mass of pyroclastic material (force of water) with those tending to resist the movement (soil strength) (Lambe and Whitman,

1979). The aforementioned approach is traditionally adopted to prevent dams failure, but it will be used, in this case, to ensure their collapse at a specific water level. During rainfall events, in fact, the barred canal section fills up rather quickly, so the hypothesis behind the simulations is that the dam reaches the instability conditions for the achievement of a given hydraulic head rather than for other processes (e.g.: erosion), since the first destabilizing condition is reached faster than the others which require longer times to be effective.

The Finite Element Method (FEM) was applied to perform the downstream slope stability analysis. The shear strength reduction (SSR) approach, which is one of the most popular techniques to perform FEM slope analysis, was adopted (Griffits et al., 1999). The SSR is simple in concept: it systematically reduces the shear strength envelope of material by a factor of safety ($FS$), and computes FEM models of the slope until deformations are unacceptably large or solutions do not converge. In classical soil mechanics, the factor of safety is the ratio of the shear strength at the plane of potential failure $\tau_f$ and the shear

stress acting in the same plane $\tau$, namely:

$$FS = \tau_f / \tau \tag{10}$$

For the Mohr–Coulomb criterion, the previous equation reads:

$$FS = (c + (s_n - u) \cdot tan\phi)/\tau \,, \tag{11}$$

where $c$ is the cohesion and $\phi$ is the friction angle of the material, $s_n$ denotes the total normal stress and $u$ the pore pressure.

For the Mohr–Coulomb model, a "reduced" set of material parameters $c^*$ and $\phi^*$ is computed:

$$c^* = c/FS_n \,, \tag{12}$$

$$tan\phi^* = tan\phi/FS_n \tag{13}$$

The problem is then solved using this set of reduced material parameters while keeping all other parameters unchanged. If convergence is obtained, the $FS_n$ value is increased and the problem is solved again. The lowest $FS_n$ producing non-convergence is reported as the "factor of safety" of the problem. If the resulting FS is greater than one, for a given water level

stressing the upstream slope of the dam, the structure is stable. If the iterative procedure gives back a unitary $FS$ value, the limit equilibrium conditions have been reached. This means that the coupling of both the dam geometrical configuration and the water level situation are going to produce the collapse of the structure. This last condition is the one that must be reached for the study purposes. The pore pressure distribution inside the dam body is a fundamental input for the strength reduction analysis. Such distribution is obtained as a function of the hydraulic head stressing the upstream slope of the embankment. The

filtration process, implemented in the finite element model, is based on the solution of the Laplace equation (Straface et al., 2010, 2011; Molinari et al., 2014; Chidichimo et al., 2015, 2018):

$$\nabla^2 h = 0 \,, \tag{14}$$

where $h(x, y)$ represents the hydraulic head distribution within the dam body which is a function of the hydraulic conductivity of the dam material. Such dependence is defined by Darcy's law:

$$q = -K\nabla h = -K\,\frac{d}{dl}\left(\frac{u}{\rho g} + z\right), \tag{15}$$

where $q$ is Darcy's velocity, $K$ is the hydraulic conductivity, $\rho$ is the water density, $u$ is the pore pressure and $z$ is the elevation above sea level. Several simulations were performed taking the parameters for the numerical model from the literature. Studies performed on the geotechnical properties of the volcanoclastic formation that is found in the Andes of Ecuador and Colombia, known as Cangahua, reported that the dry unit weight of the material was found to range around 14 kN $m^{-3}$ (Bommer et al.,

2002). The strength parameters of volcanic sediments produced by recent eruptions were investigated by Orense et al., (2006), who found a value for the friction angle ($\phi$) of these materials of about 40°, while the cohesion ($c$) is close to zero. The hydraulic conductivity ($K$) of pyroclastic beds was discovered to range between $10^{-4}$-$10^{-5}$ m $s^{-1}$ (Burgisser, 2012). A middle range value was adopted to implement the numerical models. The relatively high permeability of the pyroclastic material ensures the water outflow during the regular rainfall regimen. In case of extreme rainfall events (violent rains, typhoons, etc.),

the volcanic material is no longer able to drain the water inflow producing the water level raising which will undermine the structure stability. The dams were thought to reach a height of 3 m and to hold out up to a maximum water level of 2.6 m. Assuming the aforementioned features, the first step started from the design of a stable dam configuration (Fig. 7a). This outcome was obtained trying different dimensioning solutions in order to avoid the early collapse of the structure due, for example, to its own weight. Figure 7b shows the water velocity field in the dam cross section, while Fig. 7c shows the failure

surface in the downstream slope which is generated by a factor of safety of 1.28.

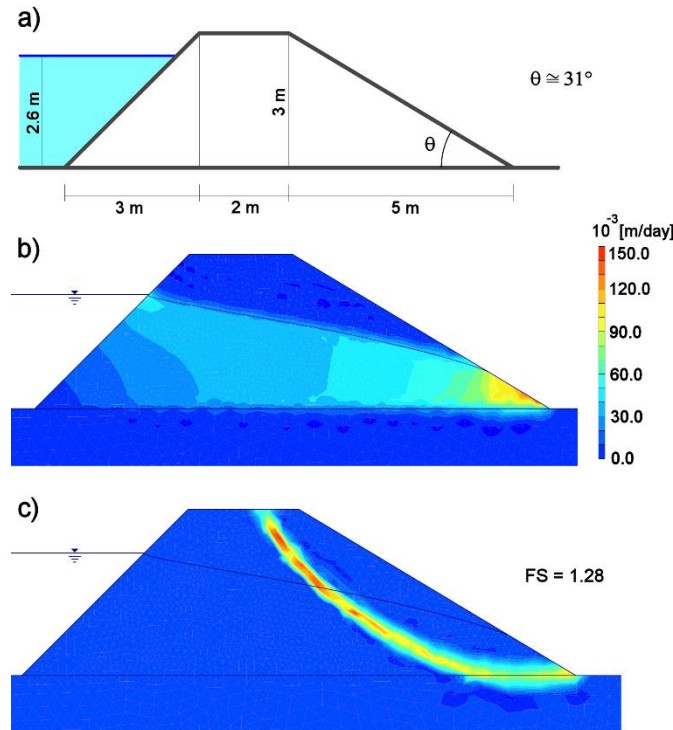

**Figure 7: - a) cross sectional sketch of a stable dam with the main elements dimension; b) water velocity field moving in the dam body and associated phreatic surface; c) failure surface generated by a factor of safety of 1.28.**

Once a stable dam was obtained for the chosen working conditions, the second step was to repeat the strength reduction analysis

5    several times by slowly increasing the inclination of the downstream slope until a unitary *FS* was reached. The inclination was

increased using the corner between the downstream slope and the dam crest as a pivot point. This resulted in a gradual increase

of the θ angle and a consequent reduction of the dam base. Figure 8 shows the analysis final result with the sizing specifications

for an easy to collapse dam built in volcanoclastic material. To ensure a greater control over the natural dam collapse timing,

a discharge channel can be arranged at the dam base. The degree of openness of this channel can be adjusted according to the

10   flow rates values observed during the extreme rainfall events recorded over time in the area, in order to delay the achievement

of the triggering hydraulic head. This ploy may be necessary to avoid the undesired simultaneous collapse of different dams;

hazard could increase when different lahars are triggered at short time intervals and reach the confluence points almost at the

same time.

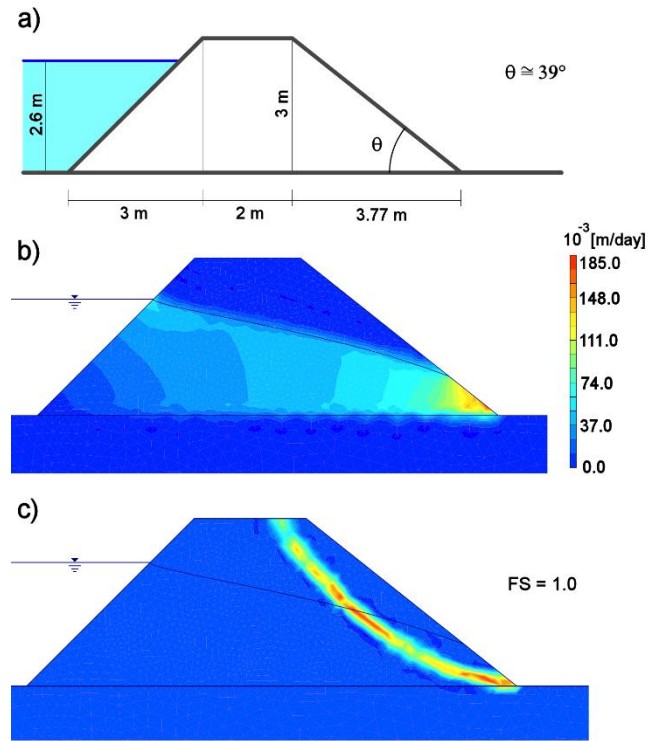

**Figure 8: a) cross sectional sketch of an instable dam with the main elements dimension; b) water velocity field moving in the dam body and associated phreatic surface; c) failure surface generated by a unitary factor of safety.**

### 4.2 Preliminary hypotheses and results of simulations

LLUNPIY was calibrated and validated for secondary lahars by simulating the two events of February 12, 2005 and August 22, 2008 occurred in the Vascún Valley of Tungurahua Volcano in Ecuador (Machado et al., 2014; Machado et al., 2015b). In particular, the 2008 event is very important in order to confirm the value of the model parameters, tuned in the simulation even where the cause of the lahar was the breakdown of a temporary pond, generated by a small landslide. Those successful simulations permitted to be confident in the scenarios which could be realized by new simulations. Of course, a very accurate

updating of geological data (DEM or DTM, detrital cover depth, etc.) and sufficient geophysical surveys are indispensable for applications aimed at lahar risk mitigation.

An initial study on the potentiality of applying mitigation measures in the Vascún Valley was performed by triggering lahars of planned size (the lahar level is here considered as the relevant datum) through the controlled collapse of rudimentary ponds. A preliminary analysis of the principal canyons of the Vascún Valley was carried out in order to individuate favorable points

for positioning embankments as dams; three points were chosen for building temporary dams: one located into Rio Vascún (point 1 in Fig. 9, Fig. 10, Fig.11 and Fig. 12), the second located in one of its tributary to the right (point 2 in Fig. 9, Fig. 10, Fig. 11 and Fig. 12) and the third in a tributary to the left (point 3 in Fig. 9, Fig. 10, Fig. 11 and Fig. 12). Rio Vascún in turn is a tributary of the much broader Pastaza River, where lahars of Vascún Valley disperse. Simulations concern the lahars

generated in the points 1, 2 and 3 (Fig. 9, Fig. 10, Fig. 11 and Fig. 12), in short, lahar 1, 2 and 3. The same initial volume of 2008 event was selected for all the simulations, except the last one. Three initial points permit to analyze an almost exhaustive set of possible conditions; we performed a sequence of simulations by LLUNPIY/3r, of course, with the same parameters values of the successful simulation of 2008 August 29 event. We present here some selected simulations, which look interesting

for many considerations, which may be deduced by their analysis.

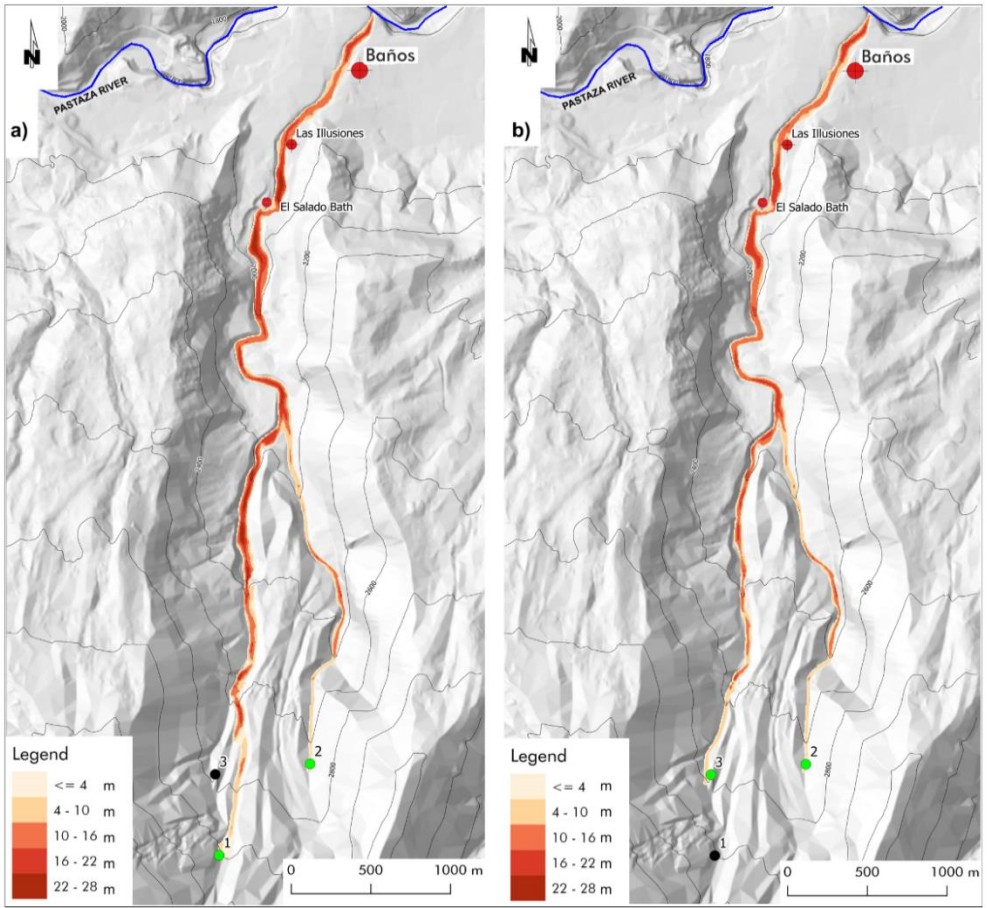

**Figure 9: Simultaneous triggering of lahars from points 1 and 2 (a), from points 2 and 3 (b). The maximum thickness of the lahar during the conjectured events is reported in meters according the legend.**

Initially, we considered two scenarios in order to investigate the effects of simultaneous and differed in time confluence of two

lahars, the results of such simulations induced us to consider a larger set of cases. The former one is generated by the simultaneous triggering from the points 1 with 4875 m$^3$ of detachment volume and 2 with 4500 m$^3$ of detachment volume (Fig. 9a), the second one is generated by the simultaneous triggering from the points 2 with the same previous detachment volume and 3 with 4250 m$^3$ of detachment volume (Fig. 9b). The confluence of the lahars in the Rio Vascún (for the following the confluence point) is almost simultaneous in the latter scenario, because of the similar distance between the triggering points

and the confluence point. The situation is diverse for the former scenario, because the distances of the triggering points from

the confluence point are very different. Intuitively the former case would be less dangerous, because the flood peaks of the two lahar cannot coincide at the confluence point, but the maximum thickness of lahar in the former scenario is 27 m., while the smaller value of 21 m. is detected in the latter scenario. The analysis of the two simulations showed that a very larger mass was eroded in the first part of the path from the point 1 in the former case. This unexpected result permits to plan an opportune

strategy according to the degree of control for triggering lahars 1, 2 and 3 from the three respective points. If triggering can be well controlled with moderate/heavy rainfall, then the best choice is to trigger lahar 3 (smaller erosion) before lahar 1, so that lahar 3 anticipates part of the erosion process in the common path of both the lahars (1 and 3) as far as the confluence point and reduces consequently the thickness of lahar 1. When the peak of lahar 3 goes beyond the confluence point, then lahar 2 can be triggered before lahar 1, which has to be generated as late as possible. Anyway, the last lahar to be triggered has to be

surely lahar 1 but a further investigation is necessary to better understand the priority between lahar 2 and 3; the study of single lahars generated in points 1, 2 and 3 could solve the question, as it may be deduced by the following simulations.

Lahar 1 causes the maximum erosion, with a maximum thickness of 26 m., because it follows the path of Rio Vascún, that is the main rio in the valley (with a larger volume of pyroclastic cover to be eroded), lahar 3 shares a relevant part of the previous path and reaches a maximum thickness of 19 m., while lahar 2, whose path is shorter before its late confluence into Rio Vascún,

involves the smallest erosion (maximum thickness of 16 m). Such results solve the doubt that we put forward with the first simulations.

An "cleaning" operation of pyroclastic cover could be projected by initially triggering lahar 2, with a first mobilization of the detrital cover also for the area related to the last part of Rio Vascún from the confluence point of lahar 2 (maximum thickness 16 m, Fig. 10b). When lahar 2 dissolves into Pastaza river, lahar 3 could be triggered with a first erosion of the detrital cover

between the confluence points of lahars 2 and 3 into Rio Vascún, so that the maximum thickness does not overcome 16 m (Fig.10c). The most dangerous lahar 1of Rio Vascún could then start at the exhaustion of lahar 3, minimizing the hazard; the maximum thickness before the confluence of lahar 3 into Rio Vascún does not overcome 22 m (Fig. 9a and Fig. 10a).

We tested successfully the outcomes of this strategy by simulating the triggering of the three lahars in successive times, each one immediately after the exhaustion of the previous one; the first phase concerns lahar 2 (Fig.10b), the maximum thickness

does not overcome 14 m in the last part of the path, from north of El Salado Bath to south of Baños.

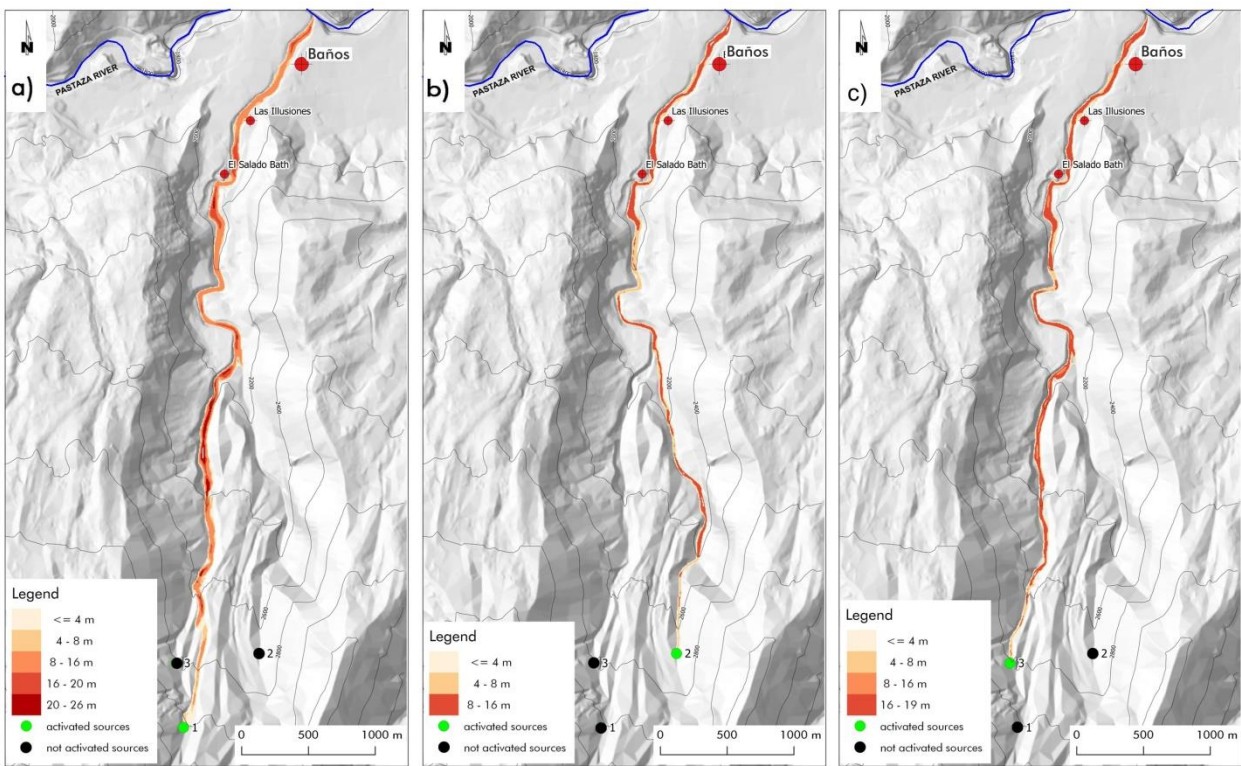

**Figure 10: Single triggerings: lahar generated in point 1 (a), in point 2 (b) and in point 3 (c). The legend specifies the maximum thickness of the lahar reached during the conjectured events for each point (cell).**

The erosion depth of pyroclastic cover, after lahar 2 exhaustion, prevents the maximum thickness of the successive lahar 3 to

overcome 10-12 m after the confluence point with lahar 2 (Fig.11a) because of the reduced pyroclastic cover, while 19 m are

reached by triggering only lahar 3 (Fig.10c).

Finally, the most dangerous lahar 1 does not overcome 4-8 m in the inhabited zones (Fig.11b), while it reaches 26 m of

maximum thickness in the last part of the path (Fig.10a), when the other lahars 2 and 3 are not generated.

This last result points out the importance of "cleaning" of pyroclastic cover according to an opportune strategy, which can be

deduced by the outcomes of simulations, which explore all the possible significant cases.

Last simulations concern two cases of simultaneous triggering of all the lahars with the same detachment volumes from points

1, 2 and 3 of the previous simulations (Fig. 12a) and the double detachment volumes from the same points (Fig. 12b) in order

to understand how triggering larger volume could increase the lahar dangerousness. The results show that the maximum

thickness of the lahar in the former case (Fig. 12a) is 28 m, while the maximum thickness of the lahar in the latter case (Fig.

12b) is 29 m, just a meter more. A double initial volume does not involve much larger erosion in this context, the joint effect

of a larger volume and erosion does not increase the hazard in a dramatic way.

The overall results confirm the goodness of the strategy of triggering lahars at different times according to an accurate analysis

of simulations after a precise knowledge of the geological features of the area of application. We remember that these

simulations were obtained without sufficient data about the pyroclastic cover (it is obviously overestimated), that would have led to more accurate results, anyway, this issue does not compromise the reliability and validity of the proposed methodology.

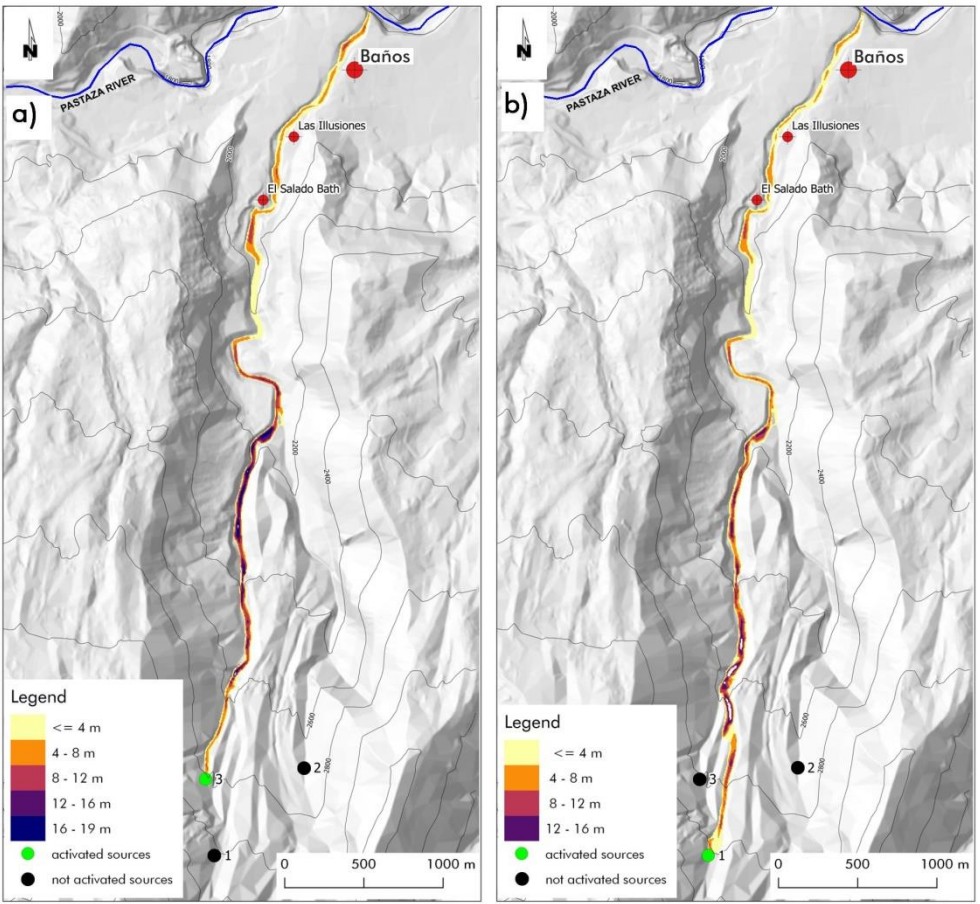

**Figure 11: Simulations of deferred triggering of lahars generated in point 3 (a), and 1(b). The legend specifies the maximum thickness of the lahar reached during the conjectured events for each point (cell).**

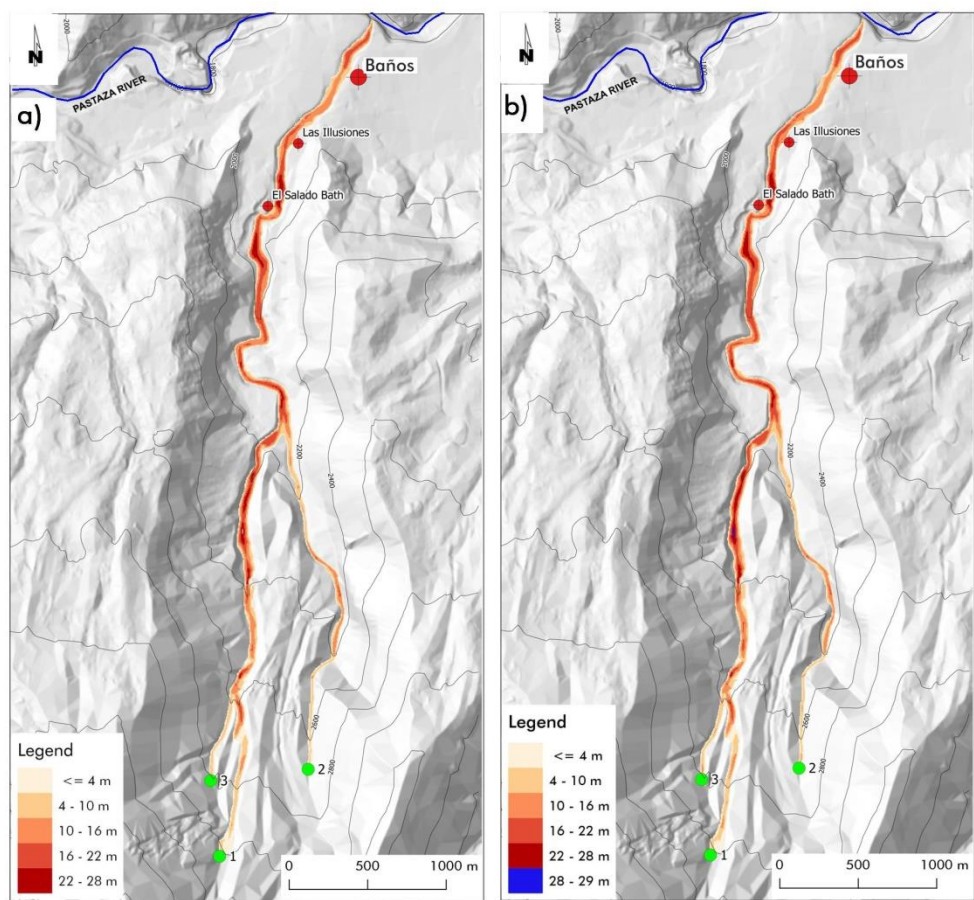

**Figure 12: Simulations of simultaneous triggering of lahars generated in point 1, 2, 3. In (a) with total detachment volume of 13625 m3 and in (b) with total detachment volume of 27250 m3. The legend specifies the maximum thickness of the lahar reached during the conjectured events for each point (cell).**

## 5 Conclusions and comments

Our case study starts from examination of some significant natural events in the Vascún Valley, an area that is heavily exposed to lahar risk. Ponds form along watercourses (rios) in the canyons of this volcanic zone, when landslides of volcanic deposits impede the normal flow. The most frequent cause of breakout of such natural ponds is the overflow of water across the newly formed dam and subsequent erosion and rapid down cutting into the loose rock debris. By eroding the blockage and flowing out watercourse downstream, the initial surge of water will incorporate part of volcanic sediments and will generate lahars. The hazard related to these lahars depends both on the features of the temporary pond and the volcanic cover along the lahar path; a larger frequency of lahars produces smaller (shorter accumulation periods) and therefore less dangerous events.

We explored the possibility to induce artificially lahars and performed many simulations for analyzing possible different scenarios for extremely complex situations; positive results obtained from these studies permit us to settle a methodology and

encourage us to continue this investigation. We propose a controlled generation of small lahars, for risk mitigation, by the collapse of temporary ponds at different times in order to avoid the superposition of lahars having the same final path. Such a proposal is out of standard and is based on observations and study of favorable situations together with the usage of a robust and well validated model of simulation in order to choose the best procedures of intervention.

The computational paradigm of Cellular Automata has allowed the development a reliable model for the simulation of lahars complex dynamics. Reliable simulation tools give the opportunity to test various hypotheses and to create related scenarios to be analyzed.

LLUNPIY/3r, the model that was used for lahar simulations in the Vascún valley is a reduced version of LLUNPIY, it does not account for the preliminary phase modelled by the fully extended LLUNPIY (Machado, 2015), when the mixing of the

rain water with the unconsolidated pyroclastic stratum originates the lahar, but it directly considers a "detachment area", the initial area where the lahar can be considered to start for simulations of both real and conjectured events. Since some data of the real event are missing at this stage, the simulation starting point corresponds to the first area crossed by the lahar, whose data have a good level of reliability.

Simulation results of lahars triggered by collapsed dam are oversized: the field data, relative to the depth of the unconsolidated

pyroclastic layer along the path of lahars, are known very approximately; a constant value of 5 meters was adopted, certainly not lower than the real one in any part of the lahar path, but possibly exaggerated in some other parts. Of course, it was preferred to consider, with poor data, an overestimated lahar hazard rather than an underestimated one, but in the future, better precise data of the unconsolidated pyroclastic layer can be obtained thanks to geophysical surveys. Anyway, even if the simulations of lahars triggered by collapsed dam produce over-valued hazard scenarios, the comparison among all the cases, where the

depth of the unconsolidated pyroclastic layer is overestimated in the same way, shows that the application of this methodology with accurate field data is worthy of being taken into due consideration. We remember that, in Ecuador, the two most accredited models for lahar simulation, LAHARZ (Muñoz-Salinas et al., 1998; Schilling, 1993) and TITAN2D (Sheridan et al., 2005; Williams et al., 2008) omit the erosion process. They impose the total amount of eroded pyroclastic layer at the first simulation step, while LLUNPIY/3r starts from an initial amount adding the new eroded quantity according to a step by step computation

of the erosive detachment.

LLUNPIY/3r is limited for application to Vascún valley (or similar cases) because all the lahars end to Pastaza river without significant variation of viscosity, so the possible last phase affecting lahars in areas with small slopes, i.e. the water loss and the resulting solidification, fails to be considered; LLUNPIY (Machado, 2015) models such a situation, but a reliable validation of the model needs simulation of opportune real cases with detailed field data.

The possibility to simulate different scenarios with reliable field data permits to forecast the thickness of lahars, their velocity, times of their peaks, to operate the best choice as potential hazard with more efficient and reliable alert procedures. Applications of LLUNPIY/3r need a thorough geological study of the area of interest, especially regarding morphology (DEM and DTM), pyroclastic soil cover, the composition of the erodible layer, also specified by geophysical surveys at the strategic

points. Furthermore, it is also important to conduct a hydrological study of watercourses, where most likely the lahars are channeled.

Feasibility studies confirmed the previous hypotheses of building weak dams with a control drainage canal with significant cost containment. Unexpected (and sometime dangerous) situations were evidenced by simulation results, which permit to evaluate the hazard of possible choices. Furthermore, more efficient and efficacious early warning protocols may be produced in such a context, social impact for partial evacuation could be mitigated. Interventions, that solve provisionally the lahar hazard, but involve future risks, can be avoided. The complexity of the objective presupposes a multidisciplinary (or perhaps transdisciplinary) approach, which implies an even greater effort to ensure those competences of different types (which are reflected by the various scientific extractions of the authors): geology, physics, mathematics, engineering, computer science can cooperate in achieving common goals. Protocols for mitigating the lahar risk can be developed in such a context, involving also social and political sciences (Leung et al., 2003, Mercer & al.,2010).

A pilot project in which the setting-up of an artificial pond is planned to conduct experiments for the triggering of small lahars in safe conditions would be necessary. This is a preliminary step for standard applications of this new strategy for reducing lahars risk.

A further achievement is the extension of LLUNPIY for modelling the flows in the urbanized area as in the last versions of SCIDDICA (Lupiano et al., 2016, 2017).

## Acknowledgements

This paper reports partly researches of the international project "Modelización y Simulación de Lahares con Autómatas Celulares mediante Computación Paralela" of University of Chimborazo. An indirect financial support was supplied by the University of Calabria.

The main contributions to this research have to be attributed to Francesco Chidichimo and Valeria Lupiano, that carried out the research and wrote the manuscript. The remaining authors revised the manuscript and the computation, provided supplementary data and validation and contributed to the editing.

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

**Table 1 - Physical and empirical parameters**

| Denotation | Description |
| --- | --- |
| $p_r$ | cell **r**adius [m] (half the distance between the center of the central cell and the center of one of its adjacent neighbors) |
| $p_t$ | **t**ime corresponding to a MCA step [s] |
| $p_{cf}$ | **c**oefficient of **f**riction [-] |
| $p_{dt,}$ | energy **d**issipation due to **t**urbulence [-] |
| $p_{pe}, p_{de}, p_{tm}$ | **p**rogressive **e**rosion [-], energy **d**issipation due to **e**rosion [-], **t**hreshold of **m**obilization [m] |
| $p_{Madh}, p_{madh}$ | **M**ax and **m**in **adh**erence [m] |
| $p_{khl}$ | loss of kinetic head [m] |

**Table 2 - Substates**

| Substates | Description |
|---|---|
| $Q_A$, $Q_D$, | **A**ltitude, pyroclastic stratum **D**epth; |
| $Q_{LT}$, $Q_{KH}$ | Lahar Thickness, Lahar Kinetic Head, |
| $Q_X$, $Q_Y$ | the co-ordinates X and Y of the lahar center of mass inside the cell |
| $Q_E$, $Q_{EX}$, $Q_{EY}$, $Q_{KHE}$ (six components) | External flow normalized to a thickness, External flow co-ordinates X and Y of mass center, Kinetic Head of External flow |
| $Q_I$, $Q_{IX}$, $Q_{IY}$, $Q_{KHI}$ (six components) | Internal flow normalized to a thickness, Internal flow co-ordinates X and Y of mass center, Kinetic Head of Internal flow |

**Table 3 - Comparison among field data, Titan2D and LLUNPIY simulation data.**

| | Field data | Simulations output Titan2D | Simulation output LLUNPIY/r3 |
|---|---|---|---|
| Mean velocity between Seismic Station and AFM | 7 m/s | - | - |
| Mean Velocity between AFM and El Salado | 3.10 m/s | - | - |
| Velocity at El Salado | 3.1m/s | 5.8–8.9. m/s | 3.1 m/s |
| Velocity at final point (Las Ilusiones) | - | 1.1–2.6 m/s | 3 m/s |
| Time between AFM station and El Salado | 16' | - | - |
| Time between start point and El Salado | - | - | 6-7' |
| Time between El Salado and Las Ilusiones | - | - | 14' |
| Total time between start point and Las Ilusiones | - | ~8-14' | 20' |
| Eroded debris between start point and El Salado | - | - | 38000 m$^3$ |
| Eroded debris between El Salado and Las Ilusiones | - | - | 71000m$^3$ |
| Total lahar volume between start point and Las Ilusiones | 55000/70000m$^3$ | 50000/70000m$^3$ | 109000 m$^3$ |

**Table 4 - Comparison between field and LLUNPIY simulation data**

|  | Field data | LLUNPIY output |
|---|---|---|
| Maximum velocity | 15 m/s | 20 m/s |
| Velocity at El Salado | 4.7 m/s | 6 m/s |
| Time between start point and El Salado bath | 5' | 4' 50" |
| Maximum flow between start point and El Salado | 640 m$^3$/s | 633 m$^3$/s |
| Total time between start point and Rio Pastaza | - | 9' |
| Total eroded debris | - | 970000m$^3$ |