# Peer review of "From examination of natural events a proposal for risk mitigation of lahars by a cellular automata methodology: a case study for Vascún valley, Ecuador"

_Natural Hazards and Earth System Sciences, 2018_

## Short Comment (SC1) · 21 Mar 2019

The paper "From examination of natural events a proposal for risk mitigation of lahars by a cellular automata methodology: a case study for Vascún valley, Ecuador" discusses hazard scenarios involving the presence of Lahars, erosive floods mixing water and pyroclastic deposits, taking advantage from LLUNPIY, a software based on the Macroscopic Cellular Automata (MCA) framework. The authors introduces the Lahars theme and discuss in detail the methodology used (i) to obtain the needed initial conditions from the huge but heterogeneous mass of data, (ii) to obtain the correct parameter

values and (iii) to evaluate strategies aiming to lower the lahars dangerousness.

The work is professional and well written, it clearly establish its main points, the background is well presented and the methods are adequate to the proposed researches. The overall presentation is well structured, clear and easy to understand by a wide and general audience, the technical language is precise and understandable.

In my opinion the main original parts of the papers resides on the idea of inducing artificially lahars (and performing many simulations for analyzing possible different scenarios in such complex situations) in order to avoid the superposition of different lahars with the same final path in natural events, reducing in such a way their gravity. It is less clear if the authors – in order to achieve such goal – added some suitable feature to the already existing MCA framework (in positive case, the authors should better highlight these new original parts)

---

## Author Comment (AC1) · 23 Mar 2019

Thanks for your careful comment! About your question regarding the presence of new original parts in the MCA model, we only summarized the parts of the overall model (it includes the primary lahars), that were used in the simulations. Software had to be modified in order to permit starting of lahars at different times and places. This point may be shortly introduced in the MCA model as an "external influence" and will be added in the correct version of the paper, opportunely highlighted. Our best regards.

---

## Short Comment (SC3) · 26 Apr 2019

The paper demonstrates how the Cellular Automata based model can be applied to simulate the phenomenon of lahars. Such investigations are very important from the point of people safety because it allows to asses the risks related to this phenomena. The authors of this paper are members of the team that for years develop models of surface flow using the Cellular Automata methodology. In this paper, the new model is also presented with high quality. It demonstrates that authors have a deep understanding of modeled phenomena. The definition of the model is followed by its application

in scenarios reflecting potentially real situations. The results are presented and thoroughly discussed. In my opinion, this paper can be accepted and published as is.

---

## Referee Comment (RC1) · Anonymous Referee #1 · 13 Jul 2019

The paper describes the use of LLUNPIY cellular automata code for the simulation of lahars occurred in 2008 in Vascun valley (Ecuador). The presented case study and the approach are interesting, but many issues hamper the publication of this manuscript as it is. The most critical point is the language. A complete revision of the text done by an English native speaker is mandatory for adapting this work to an international standard. After a total revision of the language, many scientific issues can be better verified. At this stage, it is not easy to understand if there are several scientific questions and problems that should be solved or verified or if it is only the language that hampers a real

comprehension of authors approach and results. In the following, some comments and suggestions: Page 1 – Abstract: please, revise the language and adapt the abstract to the final version of the manuscript. Sentences like: "such that equilibrium conditions could lack far" are not easily understandable by readers. Page 1 first line of introduction: published works should support this sentence. Page 2, line 11: what kind of threshold? Page 3 line 1: the definition of "external influences" is not understandable Page 3, line 5: phenomenology or phenomenon? Page 3 line 6: again, the definition of "substate" is hard to be understood. Page 3 line 15: "are our top models". This is an autoreferential approach and not a scientific approach. Page 3 line 18: please check the difference between safety and security. I think that the authors wanted to describe safety issues and not security problems. Page 3 line 20 "safety measures can increase the disaster risk in several conditions" this sentence is hard to believe. If it is true, the authors should explain better when and why. Page 3 line 25: risk reduction structures have to be maintained to be efficient. It is clear that, without maintenance, risk reduction infrastructures can create or increase the level of risk. But the problem is not the presence of infrastructures, the problem, often, is the lack of management of them. Page 4, first paragraph: I am not sure that this part should be in the introduction Page 5 line 6: what kind of fresh material? Page 4 line 13: AFM and IGENP activities should be better described also using citations. Chapter 3: the description of LLUNPIY is too limited and it is hard to understand. Some citations are used by authors to describe the LLUNPIY but readers should be able to understand this article even without reading all other cited papers. Chapter 3.2: this part should be rewritten. It is quite impossible to understand this chapter. Can pyroclastic cover mobilization and effect of turbulence considered external influences? If yes, a list of external influences should be presented, and then every element can be described. Chapter 3.3: it not clear if authors considered and simulated 2005 and 2008 lahar event or only (as presented in the title) 2008. Page 10 line 26: considering 5 meters of pyroclastic stratum seems to be a strong approximation. Have the authors considered other thickness to evaluate the impact of this approximation over the final results? Page 11 figure 5: the Authors

describe the level of performance of their simulation, but there is not a real presentation of the difference between the real event and the simulated one. In a scientific manuscript, an evaluation of the performance of the code should be supported by data and not a simple opinion of the authors. Chapter 4 Lahar triggering and effects. This chapter introduces a FEM simulation for slope stability. Readers have to read twelve pages before knowing that authors can also present a FEM analysis. If authors want to use a FEM model, they have to describe the workflow of their activity and make a better description of their research processes and results. A general comment on chapter 4: I am not familiar with pyroclastic deposits, but I had seen many streams affected by debris flow. Slope failures that caused temporary dams are often irregular and heterogeneous deposits. For this reason, if authors want to make a simulation of the temporary dam, the use of a typical geometrical section of an artificial dam seems to be not appropriate. If they want to use this geometry, they have to present better this assumption using filed data and other information. Another important point is the evolution of temporary dams. Many times, the dam break is due to the flow of the water, which fill the small temporary basin and start to flow on the dam deposit. The erosion caused by this process can create an additional destabilizing process that in this simulation, with a static level of the water has not been considered. Chapter 4.2 authors made many hypotheses, but the quality of field data seems to be very limited. That means that many of the proposed hypothesis cannot be really supported by field data. This part is interesting, but authors have to consider the introduction of a validation procedure and a numerical (and objective) evaluation of simulation performances. There is not a real discussion on this manuscript, and conclusions should be rewritten according to the improved version of the text.

---

## Referee Comment (RC2) · Anonymous Referee #2 · 25 Jul 2019

In this manuscript the authors present the model LLUNPIY (Lahar modelling by Local rules based on an UNderlying PIck of Yoked processes) following a cellular automata methodology. After describing the model, the case study of Vascún valley (Ecuador) is presented. In my opinion the cellular automata method, in reproducing lahars, is very interesting and it represents an alternative compared to "classical" approaches (Eulerian one based on conservation of mass and momentum, or Lagrangian one, i.e., particle-based method). However, before the publication, the authors should clarify some points:

[Figure]

1) The description of the model is rather brief! In table 1 the physical and empirical parameters are described, but their unit of measurement and values are missing. Then, how are these parameters calculated? The substates should be describe more extensively and clearly (for example $Q_{TH}$ in equation 1 is not indicated in table 2). In equation 3, the effect of turbulence is considered. The latter is specified by means of the Reynolds number that depends on viscosity. I am surprised to not see in cinematic equation 8 a viscosity term depending on velocity. Could you clarify this point? Finally, what are the initial conditions to start a simulation?

2) The sections regarding the simulations (3.3 and 4.2) need some improvements. It seems that the model is calibrated on 2008 event and then validated with secondary lahar events of February 2005 and August 2008, but only the latter is shown. In table 3 some field data are compared, but regarding only the 2008 event used for calibration if I have not misunderstood. To me, these points should be clarified.
* * *

---

## Author Comment (AC3) · 4 Sep 2019

September 04, 2019

Dear Reviewer #1,

We appreciate the efforts you have invested in our manuscript. I'm now posting the responses to all your comments on behalf of all co-authors for your kind consideration. The following is an itemized list of the comments and our response to each. Your comments are reported in bold to distinguish them from our responses, while the related

changes in the manuscript are in italic and enclosed in quotation marks.

Yours Respectfully.

Francesco Chidichimo, PhD Department of Environmental and Chemical Engineering University of Calabria. Via P. Bucci 42B, 87036 Rende (CS), (Italy) Ph: +39 0984 496573; Fax: +39 0984 496655; e-mail: francesco.chidichimo@unical.it

Please also note the supplement to this comment:
https://www.nat-hazards-earth-syst-sci-discuss.net/nhess-2018-406/nhess-2018-406-AC3-supplement.pdf
* * *
[Figure]

**Supplement:**

**Replies to REVIEWER #1**

**The paper describes the use of LLUNPIY cellular automata code for the simulation of lahars occurred in 2008 in Vascun valley (Ecuador).**

This paper, in the new revised form, describes the cellular model LLUNPIY for the simulation of lahars, that was validated by simulations of the 2005 and 2008 lahars of the Vascún valley (Ecuador). Both case studies have been reported in the paper. A proposal for risk mitigation of lahars, supported by LLUNPIY simulations was formulated, it provides for triggering small lahars by the controlled collapse of "ad hoc" projected temporary ponds, avoiding the formation of larger and far more dangerous lahars.

**The presented case study and the approach are interesting, but many issues hamper the publication of this manuscript as it is. The most critical point is the language. A complete revision of the text done by an English native speaker is mandatory for adapting this work to an international standard.**

We thank the anonymous reviewer for the useful comments, that help us to improve decidedly the paper. We would like to consider the problem of self-plagiarism: the authors wrote more than one hundred of papers about surface flows M&S (Modelling & Simulation) with Cellular Automata (CA), so it became difficult to find new, original but still clear expressions for both generic CA introduction and model definitions. The first version of the paper did not pass the self-plagiarism check performed by the automatic system of the journal, and the variations required to fulfill this step worsened it especially for the use of not completely appropriate synonymous. Furthermore avoiding self-plagiarism forced us to change, sometime ridiculously, denotations of substates and parameters to be used in the formulae.

**After a total revision of the language, many scientific issues can be better verified. At this stage, it is not easy to understand if there are several scientific questions and problems that should be solved or verified or if it is only the language that hampers a real comprehension of authors approach and results.**

English has been revised and corrected throughout the work, the text has been carefully reworked in the critical points preferring a more technical cut in the language in order to avoid ambiguity, but introducing some exemplification.

**In the following, some comments and suggestions:**

**Page 1 – Abstract: please, revise the language and adapt the abstract to the final version of the manuscript. Sentences like: "such that equilibrium conditions could lack far" are not easily understandable by readers.**

What in the manuscript was written as:

"*Such solutions could involve a strong environmental impact for the works and the continuous accumulation of volcanic deposits, such that equilibrium conditions could lack far, triggering more disastrous events*"

It has been replaced in the revised version by:

*"More disastrous event could be generated for the difficulty of maintaining these works in efficiency and for the changed risk conditions originating from their presence and the effects of their functioning."*

**Page 1 first line of introduction: published works should support this sentence.**

The following references have been included in the sentence:

*"Lahars are one of the most devastating phenomena as amount of fatalities in volcanic areas (Neall, 1976; Waythomas, 2014)."*

*Neall, V. E. (1976). Lahars as major geological hazards. Bulletin of the International Association of Engineering Geology-Bulletin de l'Association Internationale de Géologie de l'Ingénieur, 13(1), 233-240.*

*Waythomas, C. F. (2014). Water, ice and mud: lahars and lahar hazards at ice-and snow-clad volcanoes. Geology Today, 30(1), 34-39.*

**Page 2, line 11: what kind of threshold?**

of water height

*"if superficial water amount overcomes a threshold of water height, related to features of pyroclastic stratum and soil slope, then the percolation can cause a detachment in the unconsolidated stratum"*

**Page 3 line 1: the definition of "external influences" is not understandable**

the text has been carefully reworked in this critical point

*"The last extension of MCA are the "external influences", that account for kinds of input from the "external world" independent of local interactions (that cannot be reduced to local interactions) on some cells of the CA, e.g., the external influence "lava alimentation at the vents" is applied at each step only to the cells that correspond to vents, the value of the substate "lava quantity" is updated by adding to the previous value the lava quantity, that is considered to be discharged (in the case of simulation of a real event) in the cell during the time step or that is supposed to be discharged (in the case of simulation of a conjectured event) in the cell during the time step (Di Gregorio and Serra, 1999)."*

**Page 3, line 5: phenomenology or phenomenon?**

We apologize: phenomenology (from Greek φαινόμενον and λόγος) in English is referred only to a branch of philosophy; "*phenomenology*" was substituted by "*the typology of the phenomenon*"

**Page 3 line 6: again, the definition of "substate" is hard to be understood.**

the text has been carefully reworked in this critical point

*"Each characteristic, relevant to the evolution of the system and relative to the space portion corresponding to the cell, is individuated as a substate; the finite set Q of the states is given by the Cartesian product of the sets of substates: $Q=Q_1 \times Q_2 \times ...... \times Q_n$ , e.g., some substates for a lahar model are the average altitude of the part of territory corresponding to the cell (substate altitude), the thickness of the lahar inside the "cell" (substate lahar thickness), the depth of erodible*

*(unconsolidated) pyroclastic stratum of "soil of the cell" (substate pyroclastic stratum depth); the dynamics of the phenomenon is expressed by the variation of values of the substates for each cell in the successive steps of simulation. Note that features related to the third dimension may be expressed in terms of substates, it permits to develop two dimensions models, operating three-dimensionally in fact (Avolio et al., 2012)."*

**Page 3 line 15: "are our top models". This is an autoreferential approach and not a scientific approach.**

in order to avoid misunderstanding, the sentence "*are our top models*" was substituted by: "*are our most advanced models (in the sense that they include all the features of the previous models plus other new ones)*"

**Page 3 line 18: please check the difference between safety and security. I think that the authors wanted to describe safety issues and not security problems.**

yes, the term security was inappropriately used to replace the term safety (II version of the paper) in order to solve the self-plagiarism issue; anyway "*to organize security measures*" was substituted by "*to develop mitigation strategies*"

**Page 3 line 20 "safety measures can increase the disaster risk in several conditions" this sentence is hard to believe. If it is true, the authors should explain better when and why.**

this is almost certainly unclear as there are no specifications, which were inserted later in this introduction, in the meantime the sentence "*safety measures can increase the disaster risk in several conditions*" has been replaced by "*e.g., mitigation measures which involve engineered protection structures could modify hazard conditions in the time and could increase the disaster risk as better specified below.*"

This further sentence: "*This solution could involve a strong environmental impact: both for the works and the continuous accumulation of volcanic deposits, such that equilibrium conditions could lack far, triggering more disastrous events (Janda et al., 1981, 1996; Scott, 1989; Procter et al., 2010)*"

was replaced by "*This solution could involve a strong environmental impact, both for the difficulty of maintaining these works in efficiency, and for the changed conditions of risk originating precisely from the presence and effects of the functioning of these works (Janda et al., 1981, 1996; Scott, 1989; Procter et al., 201, Shreve & Kelman, 2014, Wisner et al., 2012).*"

Further references were added: Shreve & Kelman, 2014, Wisner et al., 2012.

Then this sentence was added: "*More in general a short paper of Kelman (2007) evidences synthetically that "Despite decades of evidence from research and practice demonstrating that reliance on structural approaches increases disaster risk over the long-term, structural approaches are frequently preferred without properly considering complementary or alternative measures. Examples of structural approaches are walls, dams, dykes, levees, and reservoirs. While they do provide some benefits, decisions to implement them and nothing else are usually made by emphasizing the short-term benefits and discounting the long-term costs.*"

Some Ilan Kelman's researches were published under the aegis of UNISDR (United Nations International Strategy for Disaster Reduction).

**Page 3 line 25: risk reduction structures have to be maintained to be efficient. It is clear that, without maintenance, risk reduction infrastructures can create or increase the level of risk. But the problem is not the presence of infrastructures, the problem, often, is the lack of management of them.**

Now this concept is explicitly considered, the sentence: *"This solution could involve a strong environmental impact both for the works and the continuous accumulation of volcanic deposits, such that equilibrium conditions could lack far, triggering more disastrous events."*

was replaced by *"This solution could involve a strong environmental impact: both for the difficulty of maintaining these works in efficiency, and for the changed conditions of risk originating precisely from the presence and effects of the functioning of these works."*

**Page 4, first paragraph: I am not sure that this part should be in the introduction**

The paragraph has been moved in the chapter *"3.3 LLUNPIY calibration and validation"*:

*"3.3 LLUNPIY calibration and validation*

*We selected the 2005 and 2008 lahars of Vascún Valley respectively for LLUNPIY/3r version calibration and validation. Available data, although incomplete, of the flood phase (Machado et al., 2015b) seemed promising in order to obtain reliable simulations. In fact data of different sources were carefully compared and analyzed (Williams et al., 2008; IGEPN, 2008) in order to reconstruct as accurately as possible the two events (Machado et al., 2014a and 2014b).*

*The use of simulation tools (from the cellular automata model LLUNPIY) needs detailed field data: DEM, depth of erodible pyroclastic stratum. It implies accurate geological investigations, including geophysical surveyssubsoil tomographies; permitting to individuate points, where dams by backfills, easy to collapse, can produce momentary ponds, whose breakdown can trigger a lahar (Machado, 2015c; Chidichimo et al., 2016)."*

**Page 5 line 6: what kind of fresh material?**

cumulated unconsolidated pyroclastic matter from volcanic eruption

*".... is subordinated both to the intensity and duration of the rainfalls and the available quantity of fresh material (cumulated unconsolidated pyroclastic matter from volcanic eruption) along the slopes and within the principal canyons (Quebradas of the Rio Vascún, Juive Grade-La Pampa Valley, Achupashal Quebrada) and in a minor from other factors."*

**Page 4 line 13: AFM and IGENP activities should be better described also using citations.**

Short sentences introduce the Instituto Geofísico Escuela Politecnica Nacional, Quito, Ecuador with its survey stations around the country, it is the maximum authority for various volcanic hazards and earthquakes. Citations significant for the paper were added:

*"Precious data were supplied by the Instituto Geofísico Escuela Politecnica Nacional, Quito, Ecuador (IGEPN) and its survey stations around the country (https://www.igepn.edu.ec/), it is the maximum authority for various volcanic hazards and earthquakes. IGEPN and its Acoustic Flow Monitor (AFM) station, which monitors passing of secondary lahars, detects most of lahars (e.g.: IGEPN, 2005; 2008a; 2008b), while many others are traced by the Observatory of the Volcano Tungurahua (OVT), which is situated 13 km to the north-northwest of the crater, also with the observation contribution of local volunteers (vigias)."*

*IGEPN: Annual Review of the Activity of Tungurahua Volcano, Technical report, Instituto Geofísico, Quito, Ecuador, www.igepn.edu.ec., 2005.*

*IGEPN: Informe tecnico preliminar del aluvion del 23 de agosto en el rio Vascún, http://www.igepn.edu.ec/tungurahua-informes/tung-especiales/tung-e-2008/8833-informe-especial-tungurahua-no-18/file, 2008a.*

*IGEPN: Weekly Report from the Tungurahua Volcano Observatory (18-24 August, 2008), Technical Report 33, Instituto Geofísico, Quito, Ecuador, www.igepn.edu.ec, 2008b.*

**Chapter 3: the description of LLUNPIY is too limited and it is hard to understand. Some citations are used by authors to describe the LLUNPIY but readers should be able to understand this article even without reading all other cited papers.**

The text has been carefully reworked and extended in more critical points, in order to understand this part of the paper without reading the cited papers. Also, the new structure of the extended Introduction, which has been divided into subsections, facilitates the understanding of this chapter.

Note that the examplifications of substates anticipate some parts of the model, clarifying some of its characteristics already at this preliminary stage.

About "Some citations are used by authors to describe the LLUNPIY but readers should be able to understand this article even without reading all other cited papers", we certainly agree about this necessity, but self-plagiarism lies in ambush along with a longer length of the paper.

In particular section 1.2 reads as follows:

"*1.2 Multicomponent or Macroscopic Cellular Automata*

*CA are both a parallel computational paradigm and an archetype for modelling "complex dynamical systems", that are extended in the space and can be described evolving mainly on the base of local interactions of their constituent parts. A homogeneous CA can be seen as a d-dimensional space, partitioned in cells of uniform size, each one embedding an identical input/output computing device (a Finite State Automaton). Input for each cell is given by the states of the neighboring cells, where the neighborhood conditions are determined by a pattern invariant in time and space. At the time t=0, cells are in arbitrary states (initial conditions) and the CA evolves changing the state at discrete times simultaneously (CA step), according to the transition function σ: $S_m \rightarrow S$, where S is the finite set of the states and m is the number of the neighbouring cells (Di Gregorio & Serra, 1999).*

*A short exemplification is given by the CA Majority: a two dimensions space is divided in square cells, the cell neighborhood is given by the cell itself and the eight surrounding cells, the states are blue (0) and red (1), the transition function calculates the sum of states in the neighborhood, if the sum is more than 4 (the majority of neighbors is red), the next state of the cell will be red otherwise it will be blue. The system evolves from an initial distribution of reds and blues sometime in a complex way, originating local points of expansion of colors (Toffoli, 1984).*

*When complex macroscopic dynamical systems as phenomena of "surface flows" (lahars, debris flows, snow avalanche, lava flows, and pyroclastic flows) are modelled by CA, the previous definitions are insufficient, Multicomponent or Macroscopic CA (MCA) adopt the following extensions.*

*The abstract CA must be related univocally to the real phenomenon in its dynamics, each cell has to correspond to a portion of the space or surface (of the territory T) where the phenomenon evolves, so the time corresponding to a step of the transition function has to be fixed, the size of the cell has to be specified e.g., by the length of its edge, these constant values in time and space are called global parameters; P is the set of global parameters, it includes both physical and empirical*

*parameters. The choice of some parameters is imposed by the desired precision of simulation where possible, e.g. cell dimension; the value of some parameters is deduced by physical features of the phenomenon, e.g. the parameter related to energy dissipation by turbulence: an initial physically sounding value is considered at the beginning of validation, such value is corrected in the phase of model validation on the base of the simulation quality by attempts, depending on comparison of discrepancies between real event and simulation results. A methodology, based on Genetic Algorithms, was usually used for calibrating the parameters of our CA models (Iovine et al., 2005).*

*Each characteristic, relevant to the evolution of the system and relative to the space portion corresponding to the cell, is individuated as a substate; the finite set Q of the states is given by the Cartesian product of the sets of substates: $Q=Q_1 \times Q_2 \times ...... \times Q_n$ , e.g., some substates for a lahar model are the average altitude of the part of territory corresponding to the cell (substate altitude), the thickness of the lahar inside the "cell" (substate lahar thickness), the depth of erodible (unconsolidated) pyroclastic stratum of "soil of the cell" (substate pyroclastic stratum depth); the dynamics of the phenomenon is expressed by the variation of values of the substates for each cell in the successive steps of simulation. Note that features related to the third dimension may be expressed in terms of substates, it permits to develop two dimensions models, operating three-dimensionally in fact (Avolio et al., 2012).*

*MCA have to account for phenomena, whose dynamics involves more interacting processes, sometime of different nature, e.g., loss of lahar energy because of erosion of the unconsolidated pyroclastic stratum of the "cell", loss of energy of the lahar in the "cell" caused by its turbolence. These interacting processes are called "elementary" processes of the CA and compose the transition function. This implies that the transition function has to be divided in parts, the "elementary processes", that are computed sequentially, each one involves the updating of the MCA substates.*

*The last extension of MCA are the "external influences", that account for kinds of input from the "external world" independent of local interactions (that cannot be reduced to local interactions) on some cells of the CA, e.g., the external influence "lava alimentation at the vents" is applied at each step only to the cells that correspond to vents, the value of the substate "lava quantity" is updated by adding to the previous value the lava quantity, that is considered to be discharged (in the case of simulation of a real event) in the cell during the time step or that is supposed to be discharged (in the case of simulation of a conjectured event) in the cell during the time step (Di Gregorio and Serra, 1999).*

*Simulations of flow-like landslides were performed by several versions of the MCA model SCIDDICA since 1987 for both subaerial and subaqueous debris/granular/mud flows (e.g., Barca et al., 1987; Avolio et al. 2008; Mazzanti et al., 2010; Avolio et al. 2013; Lupiano et al., 2014; Lupiano et al., 2015a; Lupiano et al., 2015b; Lupiano et al., 2015c; Lupiano et al., 2017). Simulations of primary and secondary lahars were performed by the MCA model LLUNPIY (Machado et al., 2014; Machado et al., 2015a; Machado et al., 2015b; Chidichimo et al., 2016).*

*LLUNPIY, SCIDDICA-SS3 and SCIDDICA-SS2 are our most advanced models (in the sense that they include the features of the previous models plus other new ones) for simulating flow-like landslides and lahars, they permit to simulate the erosion process unlike other models, that were used in lahar simulation: LAHARZ (e.g., Schilling, 1998; Muñoz-Salinas et al., 2009), TITAN2D (e.g., Sheridan, 2005; Williams, 2008; Córdoba et al., 2014)."*

Chapter 3 reads as follows:

*"3 LLUNPIY/3r model for lahar simulation*

*LLUNPIY (Lahar modelling by Local rules based on an UNderlying PIck of Yoked processes, "llunp'iy" means flood in the Quechua language) is a model for simulating secondary and primary lahars according to MCA methodology applied to complex system, whose evolution may be mainly*

*specified in terms of local interaction. MCA features of SCIDDICA-SS3 (Avolio et al., 2013) and SCIDDICA-SS2 (Avolio et al., 2008; Lupiano et al., 2016; Lupiano et al., 2017) are inherited by LLUNPIY; LLUNPIY for secondary lahars is extensively defined in Machado et al. (2015b), here are reported only the features of the model, that were applied in the study cases (reduced version LLUNPIY/3r from SCIDDICA-SS2) so no external influence was considered, the LLUNPIY/3r, simulation starts considering data for each cell related to the altitude (value of substate altitude, see Chapter 1.2), data related to the depth of erodible pyroclastic stratum of "soil of the cell" (value of the corresponding substate, see Chapter 1.2); data related to thickness of lahar, (value of the lahar thickness substate, see Chapter 1.2).*

*A reliable reconstruction of the first phase of a real event of lahar permits to fix an "initial" moment, where it is possible to deduce the thickness of lahar in the territory, these data constitute the values of the substate "thickness of the lahar" in the first step of the simulation. In the case of simulation of the collapse of a dam, that produced a momentary pond, the thickness of lahar is deduced by the mixing of pond water with the matter of dam and part of the unconsolidated pyroclastic stratum below. Note that the lahar events in the Vascun Valley, that we simulate, don't involve the very first phase of water percolation and detachment subsequent to water inclusion (Machado 2015), because the collapse of temporary pond is abrupt; in the cases of past event, data permitted simulation of the phenomenon just in the phase of lahar. Furthermore in the simulation of real and hypothesized events, all the lahars end into the Rio Pastaza, so the last phase of lahar deposition is omitted and the viscosity of lahar may be considered constant for these particular cases."*

**Chapter 3.2: this part should be rewritten. It is quite impossible to understand this chapter.**

Chapter 3.2 was deeply reworked:

[revised manuscript text omitted]

"

**Can pyroclastic cover mobilization and effect of turbulence considered external influences? If yes, a list of external influences should be presented, and then every element can be described.**

There are not external influences in this reduced version of LLUNPIY (LLUNPIY/3r). The external influences have been described in the new section "1.2 Multicomponent or Macroscopic Cellular Automata" for completeness.

"*The last extension of MCA are the "external influences", that account for kinds of input from the "external world" independent of local interactions (that cannot be reduced to local interactions) on some cells of the CA, e.g., the external influence "lava alimentation at the vents" is applied at each step only to the cells that correspond to vents, the value of the substate "lava quantity" is updated by adding to the previous value the lava quantity, that is considered to be discharged (in the case of simulation of a real event) in the cell during the time step or that is supposed to be discharged (in the case of simulation of a conjectured event) in the cell during the time step (Di Gregorio and Serra, 1999).*"

**Chapter 3.3: it not clear if authors considered and simulated 2005 and 2008 lahar event or only (as presented in the title) 2008.**

We changed the title of the chapter and added "LLUNPIY calibration and validation", together with the simulations of 2005 and 2008 Vascún Valley lahars

"*3.3 LLUNPIY calibration and validation*

[revised manuscript text omitted]

**Page 11 figure 5: the Authors describe the level of performance of their simulation, but there is not a real presentation of the difference between the real event and the simulated one. In a scientific manuscript, an evaluation of the performance of the code should be supported by data and not a simple opinion of the authors.**

A comparison between the data observed during the real events and the results obtained by the simulation have been now reported in the revised manuscript. Such comparisons can be also found in the cited papers. Please see the answer to the comment preceding the one above

**Chapter 4 Lahar triggering and effects. This chapter introduces a FEM (Finite Element Method) simulation for slope stability. Readers have to read twelve pages before knowing that authors can also present a FEM analysis. If authors want to use a FEM model, they have to describe the workflow of their activity and make a better description of their research processes and results.**

Thank you for having caught this oversight. We have included a description of the use of the FEM model in the introduction in order to provide the reader with a global view of what has been done in the study. The following explanation is now present in the revised version of the manuscript:

"*Moreover, small landslides, forming natural dams with temporary ponds, could easily trigger lahars by collapsing because of rainfalls; it sometime happens, e.g. the IGEPN (Instituto Geofísico Escuela Politecnica Nacional, Quito, Ecuador) reported such a case of August, 23 2008 (2008a; 2008b). These extraordinary combinations of events gave birth to the idea of using the overabundant pyroclastic material, available on site, to create easy to collapse artificial dams. The dam breakdown is obtained through the appropriate sizing of the cross section of the structure which is designed to fail at the achievement of a specified water level. This goal is reached through the implementation of an ad hoc numerical model, based on the Finite Element Method (FEM), for the stability analysis of the dam slopes.*"

**A general comment on chapter 4: I am not familiar with pyroclastic deposits, but I had seen many streams affected by debris flow. Slope failures that caused temporary dams are often irregular and heterogeneous deposits. For this reason, if authors want to make a simulation of the temporary dam, the use of a typical geometrical section of an artificial dam seems to be**

**not appropriate. If they want to use this geometry, they have to present better this assumption using filed data and other information.**

Thank you for this comment, since we realized that the main idea behind temporary dams needs to be better clarified into the manuscript. The natural event, reported in August 23$^{th}$ 2008 and producing what later would become a temporary dam, just inspired us to use the overabundant pyroclastic material, available on site, to create easy to collapse artificial dams. These structures are intended to generate small and frequent lahars at those points where important accumulation of volcanic material occurs or where their purpose is to avoid simultaneous confluence with other lahars. Once these aspects have been clarified, it becomes evident that it is no longer about natural dams generated by random landslide phenomena, but it is about of ad hoc structures built using the material already available on the place. That's why the simulations have been made using the typical geometrical section of an earth-filled dam. Please see the text now included into the manuscript and copied in the previous comment.

**Another important point is the evolution of temporary dams. Many times, the dam break is due to the flow of the water, which fill the small temporary basin and start to flow on the dam deposit. The erosion caused by this process can create an additional destabilizing process that in this simulation, with a static level of the water has not been considered.**

The dams are designed to not last, but to definitely collapse once a given water level is reached in the obstructed gully. During rainfall events the barred canal section fills up rather quickly, so the hypothesis behind our simulations is that the dam reaches the instability conditions for the achievement of a fixed hydraulic head first, and for erosion processes at a later time. This last destabilizing occurrence depends on the first one, so that the higher the water level upstream of the dam, the greater the erosive effect of the water flowing in the dam deposit. Again, our hypothesis is that the first condition it is reached faster than the one generated by erosion which requires longer times to be effective, that's why our simulations were intended to determine the geometry of the dam section which ensures the collapse of the structure at a given water level. Section 4.1 states now as follows:

"*The aforementioned approach is traditionally adopted to prevent dams failure, but it will be used, in this case, to ensure their collapse at a fixed water level. During rainfall events, in fact, the barred canal section fills up rather quickly, so the hypothesis behind the simulations is that the dam reaches the instability conditions for the achievement of a fixed hydraulic head rather than for other processes (e.g.: erosion), since the first destabilizing condition is reached faster than the others which require longer times to be effective.*"

**Chapter 4.2 authors made many hypotheses, but the quality of field data seems to be very limited. That means that many of the proposed hypothesis cannot be really supported by field data. This part is interesting, but authors have to consider the introduction of a validation procedure and a numerical (and objective) evaluation of simulation performances.**

As stated in the answer to the comment regarding Chapter 3.3, a calibration and validation procedure of the model, by means of the field data collected during 2005 and 2008 Vascún Valley lahars, has been added in the revised manuscript. Moreover, a discussion of the limits accompanying the model and its results is now present in the manuscript as shown in the answer to the following comment.

**There is not a real discussion on this manuscript, and conclusions should be rewritten according to the improved version of the text.**

The section "Conclusions and comments" has been expanded with a dense discussion about the limits of the version LLUNPIY/3r in its application and a short comparison with other models.

*"LLUNPIY/3r, the model that was used for lahar simulations in the Vascún valley is a reduced version of LLUNPIY, it doesn't account for the preliminary phase modelled by the fully extended LLUNPIY (Machado, 2015), when the mixing of the rain water with the unconsolidated pyroclastic stratum originates the lahar, but it considers directly a "detachment area", the initial area where the lahar can be considered to start for simulations of both real and conjectured events. When certain data of real event are missing at this stage, we consider as starting point of simulation, the first area crossed by the lahar, whose data have a good level of reliability.*

*Simulation results of lahars triggered by collapsed dam are oversized: the field data, relative to the depth of the unconsolidated pyroclastic layer along the path of lahars, are known very approximately; a constant value of 5 meters was adopted, certainly not lower than the real one in any part of the lahar path, but possibly exaggerated in some parts of the path. Of course, it was preferred to consider, with poor data, an overestimated lahar hazard rather than an underestimated one, but in the future, better precise data of the unconsolidated pyroclastic layer can be obtained thanks to geophysical surveys. Anyway, even if the simulations of lahars triggered by collapsed dam produce over-valued hazard scenarios, the comparison among all the cases, where the depth of the unconsolidated pyroclastic layer is overestimated in the same way, shows that the application of this methodology with accurate field data is worthy of being taken into due consideration. We remember that the two most accredited models in Ecuador for lahar simulation, LAHARZ (Muñoz-Salinas et al., 1998; Schilling, 1993) and TITAN2D (Sheridan et al., 2005; Williams et al., 2008) omit the erosion process, they impose the total amount of eroded pyroclastic layer at the first simulation step, while LLUNPIY/3r starts from an initial amount adding the new eroded quantity according to a step by step computation of the erosive detachment.*

*LLUNPIY/3r is limited for application to Vascún valley (or similar cases) because all the lahars end to Pastaza river without significant variation of viscosity, so the possible last phase affecting lahars in areas with small slopes, i.e. the water loss and the resulting solidification, fails to be considered; LLUNPIY (Machado, 2015) models such a situation, but a reliable validation of the model needs simulation of opportune real cases with detailed field data.*

*The possibility to simulate different scenarios with reliable field data permits to forecast the thickness of lahars, their velocity, times of their peaks, to operate the best choice as potential hazard with more efficient and reliable alert procedures. Applications of LLUNPIY/3r need a thorough geological study of the area of interest, especially regarding morphology (DEM and DTM), pyroclastic soil cover, the composition of the erodible layer, also specified by geophysical surveys at the strategic points. Furthermore, it is also important to conduct a hydrological study of watercourses, where most likely the lahars are channeled."*

---

## Author Comment (AC4) · 4 Sep 2019

September 04, 2019

Dear Reviewer #2,

We appreciate the efforts you have invested in our manuscript. I'm now posting the responses to all your comments on behalf of all co-authors for your kind consideration. The following is an itemized list of the comments and our response to each. Your comments are reported in bold to distinguish them from our responses, while the related

changes in the manuscript are in italic and enclosed in quotation marks.

Yours Respectfully.

Francesco Chidichimo, PhD Department of Environmental and Chemical Engineering University of Calabria. Via P. Bucci 42B, 87036 Rende (CS), (Italy) Ph: +39 0984 496573; Fax: +39 0984 496655; e-mail: francesco.chidichimo@unical.it

Please also note the supplement to this comment:
https://www.nat-hazards-earth-syst-sci-discuss.net/nhess-2018-406/nhess-2018-406-AC4-supplement.pdf

**Supplement:**

**Replies to REVIEWER #2**

We thank this anonymous reviewer for the useful comments, that help us to clarify some delicate parts of the paper.

**1) The description of the model is rather brief!**

The description of the model was extended and the model was specified as LLUNPIY/3r in order to distinguish this reduced version from the complete version LLUNPIY for primary and secondary lahars, furthermore the section 1.2 "Multicomponent or Macroscopic Cellular Automata" was added in the "Introduction" in order to introduce the concepts, necessary to better understand the extended description of the model, furthermore all the exemplifications related to "substates" and "elementary processes" of MCA are directly connected to LUNPIY/3r, anticipating and therefore clarifying part of its description.

In particular section 1.2 reads as follows:

*"1.2 Multicomponent or Macroscopic Cellular Automata*

*CA are both a parallel computational paradigm and an archetype for modelling "complex dynamical systems", that are extended in the space and can be described evolving mainly on the base of local interactions of their constituent parts. A homogeneous CA can be seen as a d-dimensional space, partitioned in cells of uniform size, each one embedding an identical input/output computing device (a Finite State Automaton). Input for each cell is given by the states of the neighboring cells, where the neighborhood conditions are determined by a pattern invariant in time and space. At the time t=0, cells are in arbitrary states (initial conditions) and the CA evolves changing the state at discrete times simultaneously (CA step), according to the transition function σ: $S_m \rightarrow S$, where S is the finite set of the states and m is the number of the neighbouring cells (Di Gregorio & Serra, 1999).*

*A short exemplification is given by the CA Majority: a two dimensions space is divided in square cells, the cell neighborhood is given by the cell itself and the eight surrounding cells, the states are blue (0) and red (1), the transition function calculates the sum of states in the neighborhood, if the sum is more than 4 (the majority of neighbors is red), the next state of the cell will be red otherwise it will be blue. The system evolves from an initial distribution of reds and blues sometime in a complex way, originating local points of expansion of colors (Toffoli, 1984).*

*When complex macroscopic dynamical systems as phenomena of "surface flows" (lahars, debris flows, snow avalanche, lava flows, and pyroclastic flows) are modelled by CA, the previous definitions are insufficient, Multicomponent or Macroscopic CA (MCA) adopt the following extensions.*

*The abstract CA must be related univocally to the real phenomenon in its dynamics, each cell has to correspond to a portion of the space or surface (of the territory T) where the phenomenon evolves, so the time corresponding to a step of the transition function has to be fixed, the size of the cell has to be specified e.g., by the length of its edge, these constant values in time and space are called global parameters; P is the set of global parameters, it includes both physical and empirical parameters. The choice of some parameters is imposed by the desired precision of simulation where possible, e.g. cell dimension; the value of some parameters is deduced by physical features of the phenomenon, e.g. the parameter related to energy dissipation by turbulence: an initial physically sounding value is considered at the beginning of validation, such value is corrected in the phase of model validation on the base of the simulation quality by attempts, depending on comparison of*

*discrepancies between real event and simulation results. A methodology, based on Genetic Algorithms, was usually used for calibrating the parameters of our CA models (Iovine et al., 2005).*

*Each characteristic, relevant to the evolution of the system and relative to the space portion corresponding to the cell, is individuated as a substate; the finite set Q of the states is given by the Cartesian product of the sets of substates: $Q=Q_1 \times Q_2 \times ...... \times Q_n$ , e.g., some substates for a lahar model are the average altitude of the part of territory corresponding to the cell (substate altitude), the thickness of the lahar inside the "cell" (substate lahar thickness), the depth of erodible (unconsolidated) pyroclastic stratum of "soil of the cell" (substate pyroclastic stratum depth); the dynamics of the phenomenon is expressed by the variation of values of the substates for each cell in the successive steps of simulation. Note that features related to the third dimension may be expressed in terms of substates, it permits to develop two dimensions models, operating three-dimensionally in fact (Avolio et al., 2012).*

*MCA have to account for phenomena, whose dynamics involves more interacting processes, sometime of different nature, e.g., loss of lahar energy because of erosion of the unconsolidated pyroclastic stratum of the "cell", loss of energy of the lahar in the "cell" caused by its turbolence. These interacting processes are called "elementary" processes of the CA and compose the transition function. This implies that the transition function has to be divided in parts, the "elementary processes", that are computed sequentially, each one involves the updating of the MCA substates.*

*The last extension of MCA are the "external influences", that account for kinds of input from the "external world" independent of local interactions (that cannot be reduced to local interactions) on some cells of the CA, e.g., the external influence "lava alimentation at the vents" is applied at each step only to the cells that correspond to vents, the value of the substate "lava quantity" is updated by adding to the previous value the lava quantity, that is considered to be discharged (in the case of simulation of a real event) in the cell during the time step or that is supposed to be discharged (in the case of simulation of a conjectured event) in the cell during the time step (Di Gregorio and Serra, 1999).*

*Simulations of flow-like landslides were performed by several versions of the MCA model SCIDDICA since 1987 for both subaerial and subaqueous debris/granular/mud flows (e.g., Barca et al., 1987; Avolio et al. 2008; Mazzanti et al., 2010; Avolio et al. 2013; Lupiano et al., 2014; Lupiano et al., 2015a; Lupiano et al., 2015b; Lupiano et al., 2015c; Lupiano et al., 2017). Simulations of primary and secondary lahars were performed by the MCA model LLUNPIY (Machado et al., 2014; Machado et al., 2015a; Machado et al., 2015b; Chidichimo et al., 2016).*

*LLUNPIY, SCIDDICA-SS3 and SCIDDICA-SS2 are our most advanced models (in the sense that they include the features of the previous models plus other new ones) for simulating flow-like landslides and lahars, they permit to simulate the erosion process unlike other models, that were used in lahar simulation: LAHARZ (e.g., Schilling, 1998; Muñoz-Salinas et al., 2009), TITAN2D (e.g., Sheridan, 2005; Williams, 2008; Córdoba et al., 2014)."*

Chapter 3 reads as follows:

*"3 LLUNPIY/3r model for lahar simulation*

*LLUNPIY (Lahar modelling by Local rules based on an UNderlying PIck of Yoked processes, "llunp'iy" means flood in the Quechua language) is a model for simulating secondary and primary lahars according to MCA methodology applied to complex system, whose evolution may be mainly specified in terms of local interaction. MCA features of SCIDDICA-SS3 (Avolio et al., 2013) and SCIDDICA-SS2 (Avolio et al., 2008; Lupiano et al., 2016; Lupiano et al., 2017) are inherited by LLUNPIY; LLUNPIY for secondary lahars is extensively defined in Machado et al. (2015b), here are reported only the features of the model, that were applied in the study cases (reduced version LLUNPIY/3r from SCIDDICA-SS2) so no external influence was considered, the LLUNPIY/3r,*

*simulation starts considering data for each cell related to the altitude (value of substate altitude, see Chapter 1.2), data related to the depth of erodible pyroclastic stratum of "soil of the cell" (value of the corresponding substate, see Chapter 1.2); data related to thickness of lahar, (value of the lahar thickness substate, see Chapter 1.2).*

*A reliable reconstruction of the first phase of a real event of lahar permits to fix an "initial" moment, where it is possible to deduce the thickness of lahar in the territory, these data constitute the values of the substate "thickness of the lahar" in the first step of the simulation. In the case of simulation of the collapse of a dam, that produced a momentary pond, the thickness of lahar is deduced by the mixing of pond water with the matter of dam and part of the unconsolidated pyroclastic stratum below. Note that the lahar events in the Vascun Valley, that we simulate, don't involve the very first phase of water percolation and detachment subsequent to water inclusion (Machado 2015), because the collapse of temporary pond is abrupt; in the cases of past event, data permitted simulation of the phenomenon just in the phase of lahar. Furthermore in the simulation of real and hypothesized events, all the lahars end into the Rio Pastaza, so the last phase of lahar deposition is omitted and the viscosity of lahar may be considered constant for these particular cases."*

**In table 1 the physical and empirical parameters are described, but their unit of measurement and values are missing.**

their unit of measurement were inserted

**Table 1 - Physical and empirical parameters**

| Denotation | Description |
|---|---|
| $p_r$ | cell **r**adius [m] (half the distance between the center of the central cell and the center of one of its adjacent neighbors) |
| $p_t$ | **t**ime corresponding to a MCA step [s] |
| $p_{cf}$ | **c**oefficient of **f**riction [-] |
| $p_{dt}$, | energy **d**issipation due to **t**urbulence [-] |
| $p_{pe}$ ,$p_{de}$ , $p_{tm}$ | **p**rogressive **e**rosion [-], energy **d**issipation due to **e**rosion [-], **t**hreshold of **m**obilization [m] |
| $p_{Madh}$, $p_{madh}$ | **M**ax and **m**in **adh**erence [m] |
| $p_{khl}$ | loss of kinetic head [m] |

**Then, how are these parameters calculated?**

More specifications are added in "*Introduction*" and in "*3.1 Introduction to the LLUNPIY/3r version*"

In the Introduction:

*"The choice of some parameters is imposed by the desired precision of simulation where possible, e.g. cell dimension; the value of some parameters is deduced by physical features of the phenomenon, e.g. the parameter related to energy dissipation by turbulence: an initial physically sounding value is considered at the beginning of validation, such value is corrected in the phase of model validation on the base of the simulation quality by attempts, depending on comparison of discrepancies between real event and simulation results. A methodology, based on Genetic Algorithms, was usually used for calibrating the parameters of our CA models of surface flows (Iovine et al., 2005)."*

in 3.1 Introduction to the LLUNPIY/3r version

*"Physical parameters regard physical quantities that are used in equations of the transition function and correspond to values adopted in the implementation of the model (e.g. cell apothem, that depends on several factors, data precision, insuperable approximation limits related to specific features of the phenomenon) or values as the temporal correspondence of a CA step that must account that $p_a/p_t > v_{mx}$ where $v_{mx}$ is the maximum possible velocity of flows during the development of the phenomenon, in other words the shift of a flow in a CA step hasn't to overcome the neighborhood. All the parameters, except $p_a$ and $p_t$, are empirically fixed in the phase of model validation by the simulation quality, initial values of parameters were deduced by the physical features of the phenomenon, e.g. a very slow lahar would emerge unbelievably in simulation by largest values of $p_{cf}$, the coefficient of friction and $p_{dt}$, the energy dissipation due to turbulence".*

**The substates should be describe more extensively and clearly (for example $Q_{TH}$ in equation 1 is not indicated in table 2).**

The substates have been now described in the revised manuscript:

*"Each characteristic, relevant to the evolution of the system and relative to the space portion corresponding to the cell, is individuated as a substate; the finite set Q of the states is given by the Cartesian product of the sets of substates: $Q=Q_1 \times Q_2 \times ...... \times Q_n$ , e.g., some substates for a lahar model are the average altitude of the part of territory corresponding to the cell (substate altitude), the thickness of the lahar inside the "cell" (substate lahar thickness), the depth of erodible (unconsolidated) pyroclastic stratum of "soil of the cell" (substate pyroclastic stratum depth); the dynamics of the phenomenon is expressed by the variation of values of the substates for each cell in the successive steps of simulation. Note that features related to the third dimension may be expressed in terms of substates, it permits to develop two dimensions models, operating three-dimensionally in fact (Avolio et al., 2012)."*

$Q_{TH}$ and $Q_{LT}$ denote the same substate, all the $Q_{TH}$ were substituted by $Q_{LT}$. Thank you for catching the mistake.

*"Pyroclastic cover mobilization*
*Soil features together with the quantity of water content determine a value $p_{tm}$ of mobilization threshold to be compared with the kinetic head $Q_{KH}$ of lahar debris inside the cell, when $Q_{KH} > p_{tm}$, then the pyroclastic cover is eroded, the lahar thickness augments and altitude diminishes according to the following empirical formula, that turned out to be valid in different models of debris flow e.g. (Avolio et al., 2008), snow avalanche (Avolio et al., 2017) and primary and secondary lahars e.g. (Machado, 2015).*

$$-\Delta Q_D = \Delta Q_{LT} = -\Delta Q_A = (Q_{KH} - p_{tm})\, p_{pe} , \qquad (1)$$

*There is correspondingly a dissipation of energy, proportional to the depth of erosion, it is specified by a decrease of kinetic head according to the following formula:*

$$-\Delta Q_{KH} = (Q_{KH} - p_{tm})\, p_{de} \, , \qquad\qquad (2)\text{"}$$

**In equation 3, the effect of turbulence is considered. The latter is specified by means of the Reynolds number that depends on viscosity. I am surprised to not see in cinematic equation 8 a viscosity term depending on velocity. Could you clarify this point? Finally, what are the initial conditions to start a simulation?**

These two points were clarified in different parts of the paper: no external influence was considered in LLUNPIY/3r simulations.

*"This "adherence" method was initially used for modelling lava flows by CA, in order to manage the continuous variation of viscosity by cooling of lava e.g., (Avolio et al., 2006). The approximation for accounting for viscosity inside a CA context can be intuitively explained as follows: instead of considering innumerable layers of fluid flowing over one another, at most two layers are considered, the first layer, whose maximum thickness (adh) is determined by the coefficient of viscosity, cannot move, if the thickness of fluid  th overcomes adh, a second layer with thickness th-adh  is considered to slide on the first one with a friction coefficient related to viscosity."*

The simulation starts as follows:

*"A reliable reconstruction of the first phase of a real event of lahar permits to fix an "initial" moment, where it is possible to deduce the thickness of lahar in the territory, these data constitute the values of the substate "thickness of the lahar" in the first step of the simulation. In the case of the simulation of a lahar produced by the collapse of a dam holding a momentary pond, the thickness of lahar is deduced by the mixing of pond water with the dam material and part of the unconsolidated pyroclastic stratum below. Note that the simulated lahar events, occurred in the Vascun Valley, don't involve the very first phase of water percolation and detachment subsequent to water inclusion (Machado 2015), since the collapse of temporary pond is abrupt; in the cases of past event, data permitted simulation of the phenomenon just in the phase of lahar. Furthermore in the simulation of real and hypothesized events, all the lahars end into the Rio Pastaza, so the last phase of lahar deposition is omitted and the viscosity of lahar may be considered constant for these particular cases."*

**2) The sections regarding the simulations (3.3 and 4.2) need some improvements. It seems that the model is calibrated on 2008 event and then validated with secondary lahar events of February 2005 and August 2008, but only the latter is shown.**

Now data of both the events are reported, in particular for February 2005 event. A comparison with TITAN2D simulation is also reported.

*"3.3 LLUNPIY calibration and validation*

*We selected the 2005 and 2008 lahars of Vascún Valley respectively for LLUNPIY/3r version calibration and validation. Available data, although incomplete, of the flood phase (Machado et al., 2015b) seemed promising in order to obtain reliable simulations. In fact data of different sources were carefully compared and analyzed (Williams et al., 2008; IGEPN, 2008) in order to reconstruct as accurately as possible the two events (Machado et al., 2014a and 2014b).*

*The simulation of 2005 event is based on a Digital Elevation Model (DEM) with 1m cell size (supplied to us by Dr. Gustavo Cordoba), while the 2008 lahar was performed with a DEM of 5m*

*cell size (supplied by IGEPN). In both cases a uniform thickness of 5 m was imposed for detrital cover, because detailed surveys were not available. This introduces a series of approximations that influence negatively the results of simulations. Such approximations can be reduced by an opportune survey of field data, e.g. by soil tomographies, MASW, coring, etc.*

*The same set of LLUNPIY/3r parameters was used in the two cases except for the parameter of progressive erosion ($p_{pe}$) because of different percentages of water in the soil. The 2005 event was triggered in a higher and very slope zone of Rio Vascún, when the water concentration in the soil, by rainfall, reached critical values. The 2008 event was dissimilar, because the breaking of a temporary pond released suddenly a larger water quantity (in comparison with 2005 case) with strong turbulence, whose effects correspond to a higher value of the parameter of progressive erosion (Machado et al., 2015b).*

*The results of the simulations of 2008 event (Machado et al. 2015b) are extensively reported in this study since this event is very important because it was caused by a breaking of a temporary pond, the same typology of the 
[revised manuscript text omitted]

**In table 3 some field data are compared, but regarding only the 2008 event used for calibration if I have not misunderstood. To me, these points should be clarified.**

Please see the answer to the previous comment

---

## Author Comment (AC5) · 4 Sep 2019

Dear Pawel Topa,

we want to thank you for considering our study as a high quality work accompained by interesting and well presented results.

---

## Author Response (AR1)

October 24, 2019

To: Filippo Catani
Editor
NHESS

**Revised Manuscript nhess-2018-406**, **From examination of natural events a proposal for risk mitigation of lahars by a cellular automata methodology: a case study for Vascún valley, Ecuador.** by Valeria Lupiano, Francesco Chidichimo, Guillermo Machado, Paolo Catelan, Lorena Molina, Claudia R. Calidonna, Salvatore Straface, Gino M. Crisci, Salvatore Di Gregorio

Dear Editor,

We appreciate the constructive comments you made in this second revision which further improved our manuscript. We are now submitting a revised version for your kind consideration. In this deep revised manuscript, Dr. Claudia Roberta Calidonna has been added as new author, since she gave an important contribution to address referees' remarks since the first revision process.

The following is an itemized list of the comments and our response to each. Your comments are reported in bold to distinguish them from our responses, while the related changes in the manuscript are in italic and enclosed in quotation marks.

The new amendments are indicated through the Microsoft Word's Track Changes feature.

Yours Respectfully.

Francesco Chidichimo, PhD

Department of Environmental and Chemical Engineering

University of Calabria. Via P. Bucci 42B, 87036 Rende (CS), (Italy)

Ph: +39 0984 496573; Fax: +39 0984 496655; e-mail: francesco.chidichimo@unical.it

**Replies to EDITOR**

**Dear authors, after the latest revisions that you have introduced, I think that most of the technical and science-related issues are now solved. I believe that you have truly spent a lot of time and effort on the modifications and amendments of the manuscript and that now we can consider the peer-review part as completed.**

Thank you for the positive opinion expressed on our work, we truly spent a lot of time and efforts to improve the manuscript following the constructive comments raised by the reviewers.

**However, in the present form, the text still fails in satisfying international standards as per English writing.**

**The same thing has been noted by all referees throughout all the manuscript discussion and revision process. I acknowledge that you have gone a long step forward in ameliorating the readability of the text. Nonetheless, there are still several parts that are difficult to understand and many others where fluency is impeded by an awkward usage of terms and sentences.**

**The entire manuscript needs careful and general polishing by a fluent English speaker, to shorten and simplify unnecessary long sentences, to avoid awkward wording and to opt for standard geological and geomorphological terminology when necessary.**

English has been revised and corrected throughout the work, the text has been carefully reworked to shorten and simplify long sentences, to improve the fluency removing awkward wording and improving the readability.

**page 1 row 14 - "... they look the biggest environmental disasters..."**

the text has been reworked in this point:

*"Lahars are erosive floods, mixtures of water and pyroclastic detritus, known for being the biggest environmental disaster and causing a large number of fatalities in the volcanic areas."*

**row 19 - "... maybe for the climatic change..."**

This part has been replaced in the revised version by:

*"The growing frequency of lahars in the Vascún Valley of Tungurahua Volcano (Ecuador), probably due to the effects of the climatic change, has recently produced smaller and less dangerous events, sometimes favoured by the collapse of ponds generated by small landslides."*

**many times in the doc - wrong usage of verb "to fix" as a synonym of set.**

We apologize, the problem has been now "fixed" in the revised version of the manuscript

**pag 4 row 19 - "... must be studied from an interdisciplinar way..."**

The sentence has been reworded as follows:

*"Such instruments have to be used with extreme caution, because the complex problem of lahar hazard must be studied with an interdisciplinary approach"*

**pag 4 rows 26-27 - "...both for the difficulty of maintaining these works in efficiency, and for the changed conditions of risk originating precisely from the presence and effects of the functioning of these ..."**

the text has been reworked in this point

"*This solution could involve a strong environmental impact: it is difficult to guarantee the constant efficiency of these works, and their presence, together with the effects of their functioning could severally change the risk conditions*"

**pag 8 row 30 - "... momentary pond ..."**

This definition has been removed from the manuscript and replaced using just "*pond*" or by completely rewording the sentence as in the point specified in your comment:

"*In the case of the simulation of a lahar produced by the collapse of a dam holding a given water volume, the thickness of lahar is deduced by the mixing of pond water with the dam material and part of the unconsolidated pyroclastic stratum below.*"

**pag 11 row 4-5 - Double repeated subject in the same sentence (occurs very often in manuscript due to sentences that are too long)**

As said in the answer of the first general comment, the text has been carefully reworked to shorten and simplify long sentences, to improve the fluency removing awkward wording and improving the readability.

**many times in the doc - wrong usage of term "matter" instead of "soil", "terrain", "sediment", "material"**

The term "matter" has been substituted at all points where it was used improperly:

"*pyroclastic material*", "*pyroclastic deposits*", "*variation of lahar content*"

**Furthermore, I have noticed that readability would greatly gain from the introduction of symbols to substitute long variable names which often appear in the descriptions, such as e.g. "soil of the cell"**

Yes, long variable names have been substituted with symbols in few parts, however, in those points in which the sentence was explicative of formulae, we prefer to add symbols without cancelling the variable name in order to obtain the utmost clarity. Of course this operation was not carried out in the introduction as the model has not yet been defined in this section.

**Finally, please avoid the usage of contracted forms such as "hasn't" instead of "has not".**

All the contracted forms have been replaced by their complete forms.

[revised manuscript text omitted]

---

## Author Response (AR2)

November 13, 2019

To: Filippo Catani
Editor
NHESS

**Manuscript nhess-2018-406**, **From examination of natural events a proposal for risk mitigation of lahars by a cellular automata methodology: a case study for Vascún valley, Ecuador.** by Valeria Lupiano, Francesco Chidichimo, Guillermo Machado, Paolo Catelan, Lorena Molina, Claudia R. Calidonna, Salvatore Straface, Gino M. Crisci, Salvatore Di Gregorio

Dear Editor,

We are submitting the corrected version of the manuscript following your indications. In particular, the unwanted comment in the Conclusion section ("The main contributions to this research...."), has been removed. The author's personal contributions have been now included in the Acknowledgment section, and the contribution of all authors has been listed.

Yours Respectfully.

Francesco Chidichimo, PhD

Department of Environmental and Chemical Engineering

University of Calabria. Via P. Bucci 42B, 87036 Rende (CS), (Italy)

Ph: +39 0984 496573; Fax: +39 0984 496655; e-mail: francesco.chidichimo@unical.it